# Structural basis of hydroxycarboxylic acid receptor signaling mechanisms through ligand binding

Shota Suzuki [1], Kotaro Tanaka[2,3], Kouki Nishikawa[4], Hiroshi Suzuki [1], Atsunori Oshima [2,3,5,6] & Yoshinori Fujiyoshi [1] ✉

Hydroxycarboxylic acid receptors (HCA) are expressed in various tissues and immune cells. HCA2 and its agonist are thus important targets for treating inflammatory and metabolic disorders. Only limited information is available, however, on the active-state binding of HCAs with agonists. Here, we present cryo-EM structures of human HCA2-Gi and HCA3-Gi signaling complexes binding with multiple compounds bound. Agonists were revealed to form a salt bridge with arginine, which is conserved in the HCA family, to activate these receptors. Extracellular regions of the receptors form a lid-like structure that covers the ligand-binding pocket. Although transmembrane (TM) 6 in HCAs undergoes dynamic conformational changes, ligands do not directly interact with amino acids in TM6, suggesting that indirect signaling induces a slight shift in TM6 to activate Gi proteins. Structural analyses of agonist-bound HCA2 and HCA3 together with mutagenesis and molecular dynamics simulation provide molecular insights into HCA ligand recognition and activation mechanisms.

The hydroxycarboxylic acid receptor (HCA) family consists of the typical metabolism-sensing receptors present in humans and belongs to the class A GPCRs. The HCA family comprises 3 subtypes, HCA1 responding to lactate[1], HCA2 responding to niacin and hydroxybutyrate (BHB)[2], and HCA3 responding to 3-hydroxyoctanoic acid[3], and signals through the inhibitory Gi/o family of G proteins[4]. Downstream signaling is diverse and tissue-dependent. Among these subtypes, HCA2 is predominantly expressed in the intestines and white/brown adipocytes, as well as in various immune cells, including dendritic cells, monocytes, macrophages, neutrophils, and epidermal Langerhans cells[5–8]. Therefore, HCA2 is involved in many pathophysiologic processes. Recent studies demonstrated that potent drugs acting on HCA2 could have beneficial effects on multiple neurologic diseases. For example, an FDA-approved formulation of niacin,

Niaspan, stimulates a broad and complex protective response mediated by microglia, leading to a lower plaque burden, reduced neuronal loss, and improvements in working memory deficits[9]. BHB, a ketone body, also induces a neuroprotective phenotype in bone marrow-derived macrophages invading the brain and acts as an endogenous factor that protects against stroke and neurodegenerative diseases, and this action is mediated by HCA2[5].

Several potent HCA2 agonists, including niacin-containing acipimox, acifran, and monomethyl fumarate (MMF), are currently approved for the clinical treatment of cardiovascular and neurologic diseases. Niacin is being investigated as a treatment for Parkinson's disease[10] and glioblastoma due to its immunomodulatory and neuroprotective properties and is currently undergoing clinical trials (NCT03808961, NCT04677049)[9]. The niacin-derived antiphlogistic

[1]TMDU Advanced Research Institute, Tokyo Medical and Dental University Bunkyo-ku, Tokyo, Japan. [2]Cellular and Structural Physiology Institute (CeSPI), Nagoya University, Nagoya, Japan. [3]Department of Basic Medicinal Sciences, Graduate School of Pharmaceutical Sciences, Nagoya University, Nagoya, Japan. [4]Joint Research Course for Advanced Biomolecular Characterization, Tokyo University of Agriculture and Technology, Tokyo, Japan. [5]Institute for Glyco-core Research (iGCORE), Nagoya University, Nagoya, Japan. [6]Center for One Medicine Innovative Translational Research, Gifu University Institute for Advanced Study, Gifu City, Japan. ✉e-mail: yoshi.cesp@tmd.ac.jp

agents acipimox and acifran are commonly used clinically to treat dyslipidemia and atherosclerosis[11,12]. In addition, MMF was approved by the FDA in 2020 for the treatment of relapsing-remitting multiple sclerosis[13]. Experimental evidence shows that MMF activates HCA2, resulting in a change in microglia from a pro-inflammatory form to a neuroprotective form. Thus, HCA2 is an attractive drug target for a very diverse range of diseases, but there are several problems with the molecule. These drugs cause severe flushing (known as the niacin flush), which decreases patient compliance. Therefore, much effort has been focused on developing alternatives associated with less flushing. LUF6283, belonging to the pyrazole class of compounds, is a partial agonist with lower affinity[14]. LUF6283 achieves the action of niacin without the undesirable flushing side effects. In addition, the high-affinity HCA2-selective agonists MK-6892[15], SCH900271[16], and GSK256073[17] were developed. GSK256073 binds both HCA2 and HCA3 but is 100-fold more selective for HCA2. MK6892 is an HCA2-selective agonist with a chemical structure that differs from the 3 preceding compounds. The lack of structural information for the active state of any of the HCA subtypes as well as the lack of a structural framework for HCA-ligand binding and selectivity, however, substantially impede advances in rationale drug discovery.

HCA3 (GPR109b) exists only in hominids, including humans. Despite the high sequence similarity between HCA2 and HCA3, their endogenous ligands differ. Pharmacologic and computational analyses indicate that amino acids in the extracellular half of the transmembrane (TM) domain are responsible for the different ligand preferences[18,19]. How differences in the ligand binding pockets determine ligand preference, however, is still unclear. Structural information on HCA2 and HCA3 will provide important clues for the development of subtype-selective drugs.

In this work, we determined the cryo-EM structures of human HCA2-Gi complexes bound to the HCA2-selective full agonists GSK256073 and MK6892, the partial agonist LUF6283, and the non-selective synthetic full agonist acifran to investigate the molecular mechanisms underlying HCA ligand recognition and activation. For a deeper understanding of HCAs, we also analyze the structure of the human HCA3-Gi complex with acifran. These structural and mutational analyses and molecular dynamics (MD) simulation experiments reveal the molecular basis for understanding how HCA recognizes ligands and activates G proteins.

## Results

### Cryo-EM structures of HCA2 and HCA3–Gi complexes with different agonists

We generated HCA2 and HCA3 constructs with thermostabilized apocytochrome b562RIL (bRIL) conjugated at the N-terminus of the receptors and, for HCA2, fused the receptor to the large NanoBiT subunit (LgBiT) to stabilize the complex (NanoBiT tethering strategy[20,21]). These modifications had little effect on the pharmacologic properties of HCA2 and HCA3 (Supplementary Fig. 1). The receptor, Gi, Gβγ subunits, and scFv16 were co-expressed in Sf9 insect cells and then incubated with chemically different agonists for complex assembly, which yielded homogenous complex samples for structural studies. We obtained the cryo-EM maps of HCA2-Gi-scFv16 (HCA2-Gi) in complex with 4 agonists at overall resolutions of 2.9 Å (GSK256073), 3.0 Å (MK6892), 3.1 Å (LUF6283), and 3.2 Å (acifran), as well as a map of HCA3-Gi-scFv16 (HCA3-Gi) in complex with acifran at an overall resolution of 3.2 Å (Fig. 1, Supplementary Figs. 2–6, and Supplementary Table. 1). By local refinement covering only the TM domain on cryoSPARC[22], we obtained clear and continuous density maps of the entire receptor part, including the isolated ligand densities and extracellular regions. These maps allowed us to unambiguously build models of HCA2 and HCA3 in complex with the Gi heterotrimer. We did not observe precise densities for the first ~8 amino acids and the C-terminus (F301-), however, suggesting a disordered conformation of these regions. As in most cryo-EM structures of GPCR-G protein complexes, the α-helical domains of the Gi and NanoBiT complexes were not resolved due to their flexibility[23-25].

### HCA2 ligand binding pockets

Despite the chemical diversities of the drugs, the agonist-bound HCA2-Gi complexes had similar overall structures (Fig. 2a). All agonists, except for MK6892, had a very similar binding pose at the orthosteric site. As the ligand binding pocket of HCA2 is formed by TM1, TM2, TM3, TM7, and extracellular loop 2 (ECL2) (Supplementary Fig. 7a), the binding site differs significantly from that of well-studied class A monoamine-coupled receptors, such as serotonin[26] and dopamine receptors[27]. The ligand-binding pocket of amine-coupled receptors comprises mainly TM3, TM5, TM6, and TM7 (Supplementary Fig. 7a). In HCA2, the 3.32 position is leucine $L^{3.32}$ (superscripts denote generic Ballesteros–Weinstein numbering[28]), contributing to hydrophobic

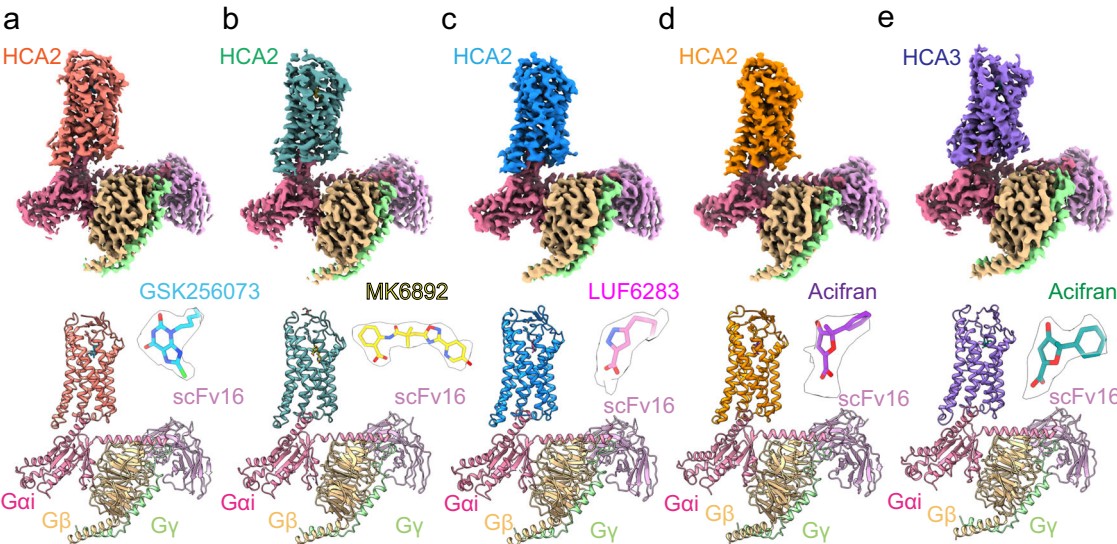

**Fig. 1 | Overall structures of the HCA2-Gi and HCA3-Gi signaling complexes. a–e** Cryo-EM maps (top panels) and the modeled structures (bottom panels) of HCA2-Gi complexes with GSK256073, MK6893, LUF6283, and acifran, and HCA3-Gi complexes with acifran. Color schemes are indicated by the labels. Ligands are shown in stick representation, and the close-up view of each ligand and the cryo-EM density are extracted and overlaid (upper right of bottom panels).

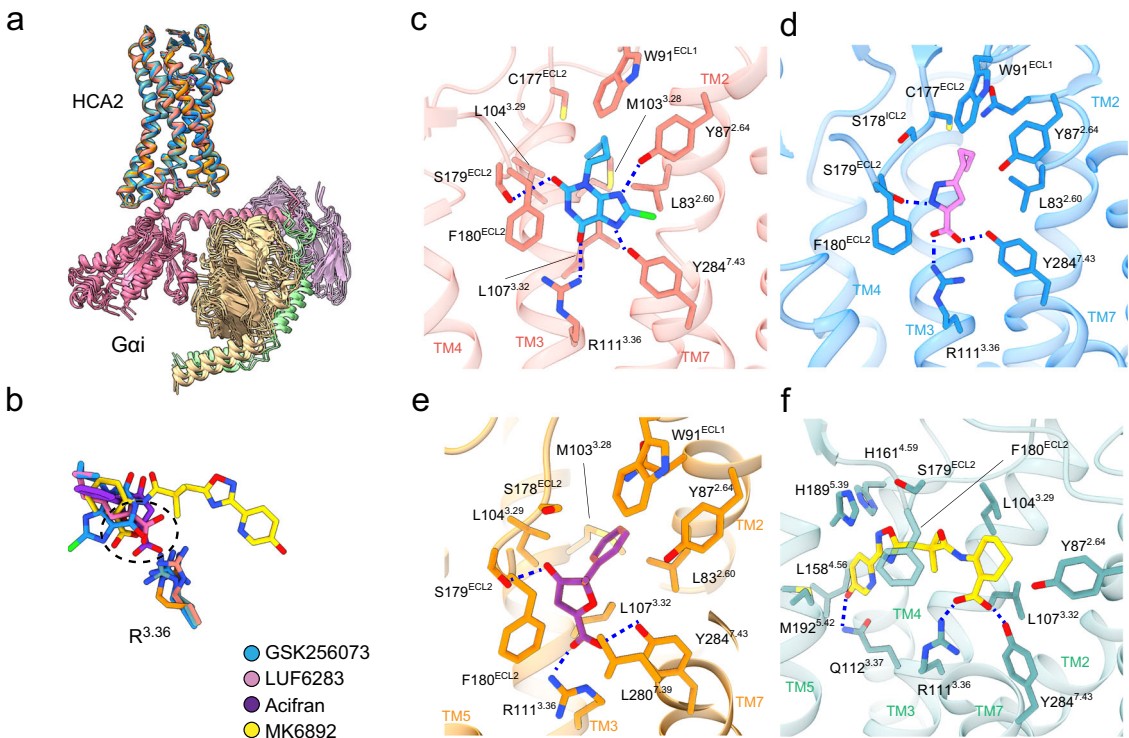

**Fig. 2 | Orthosteric binding pocket of active HCA2 binding with different agonists. a** Superposition of 4 structures of HCA2 signaling complexes aligned based on the receptor regions. **b** Superposition of GSK256073, MK6892, LUF6283, and acifran, modeled in the cryo-EM structures. The carboxyl moiety is highlighted by the dashed circle. **c-f** Detailed interactions of GSK256073 (**c**), LUF6283 (**d**), acifran (**e**), and MK6893 (**f**) with HCA2. Residues within 4 Å from the agonist are labeled and shown in stick representation. Polar interactions are indicated by blue dashed lines.

interactions and construction of the ligand-binding pockets. On the other hand, in amine-coupled receptors, the ligands interact with conserved amino acids $D^{3.32}$ (Supplementary Fig. 7b–e), whose substitution with alanine results in reduced potency[26,27,29].

The salt bridge between the ligands and $R^{3.36}$, which was observed for all agonists (Fig. 2b), is unique. Alanine substitution of $R111^{3.36}$ completely abolished the receptor activity (Fig. 2c-f, Supplementary Fig. 8a, d, g, j), while its cell surface expression level was comparable to that of the WT (Supplementary Fig. 9a). $R^{3.36}$ is conserved only in the HCA family, GPR35[30], OXER[31] and GPR31 (Supplementary Fig. 10). Although the function of $R^{3.36}$ in ligand recognition for the latter 2 GPCRs is not known, the structures of HCA2 suggest that $R^{3.36}$ is essential for HCA family recognition of the ligand. The $S179^{ECL2}$ and $Y284^{7.43}$ in HCA2 also form hydrogen bonds with the ligands (Fig. 2c-f). Alanine mutants of these amino acids reduced the activity, suggesting that $S179^{ECL2}$ and $Y284^{7.43}$ are involved in recognizing agonists and affect receptor activity (Supplementary Fig. 8h, i, k, l). The acyl tails of GSK256073 and LUF6283, and the aromatic ring of acifran are surrounded by several hydrophobic residues ($L83^{2.60}$, $W91^{ECL1}$, $M103^{3.28}$, $L107^{3.29}$, $C177^{ECL2}$, $F180^{ECL2}$, and $F276^{7.35}$), which form hydrophobic and van der Waals interactions with them, and fit into the ligand-binding pocket. Purine-2,6-dione of GSK256073 is larger than the furan ring of acifran or the pyrazole ring of LUF6283 and has a tighter contact with those residues than acifran and LUF6283, resulting in a 100-fold higher potency (Fig. 2 and Supplementary Fig. 8a, d, g, j).

Computational analysis suggested that $R251^{6.55}$ is essential for ligand recognition[18,32]. Although there is no direct interaction between $R251^{6.55}$ and any of the ligands, $R251^{6.55}$ forms hydrogen bonds with the backbone carbonyl of $S181^{ECL2}$ (Supplementary Fig. 11a). The hydrogen bond stabilizes ECL2 and may be indirectly involved in ligand binding by fixing the position of $F180^{ECL2}$, a component of the ligand binding pocket. Alternatively, the region around $R251^{6.55}$ is crowded with aromatic residues, which may contribute to stabilizing the upper part of

the ligand-binding pocket through π-cation interactions (Supplementary Fig. 11a). Consistent with the previous report, experimental mutagenesis showed that the R251A mutant exhibited an almost 100-fold reduction in receptor activity and that alanine substitutions of any of the residues in the aromatic clusters ($F180^{ECL2}$, $F193^{5.43}$, and $F276^{7.35}$) greatly decrease the activity by all agonists (Supplementary Fig. 8).

The carboxylic acid of MK6892 occupied a common position with the other 3 agonists, and the hydroxypyridine side was extended to TM4 and TM5 to form additional interactions (Fig. 2f). The hydroxypyridine of MK6892 and $Q112^{3.37}$ forms a hydrogen bond and the oxadiazole of MK6892 forms a van der Waals interaction with $H189^{5.39}$. These additional interactions explain the high affinity (EC$_{50}$ 4 ~ 20 nM) of MK6892. Mutational analysis showed that Q112A decreased the potency of only MK6892. (Supplementary Fig. 8e). Rotamers of $H189^{5.39}$ in the MK6892-bound structure were conformationally altered to fit the ligand compared with the other agonists (Supplementary Fig. 11b–e). In these structures, except for the MK6892-bound structure, S179 and H189 form hydrogen bonds, but no hydrogen bond was observed in the MK6892-bound structure and may be rather unnecessary. These observations suggest the high flexibility of the ligand-binding pocket and explain the ability of the HCA2 receptor to bind various ligands.

## Active conformation of HCAs

Structural comparison of the agonist-bound HCA2 with the antagonist-bound succinate receptor (SUCR)[33], which has high sequence similarity of microswitches and recognizes a similar ligand, succinate, allowed us to infer a mechanism of activation of HCA2 (Supplementary Fig. 12a). Comparison with the antagonist-bound SUCR structure revealed that HCA2 adopts a fully active conformation with an outward shift of the cytoplasmic side of TM6 occurring in the characteristic activation state of class A GPCRs[34,35] (Fig. 3a). The outward shift of TM6 permits insertion of the C-terminus of Gi, which would clash with TM6 in the

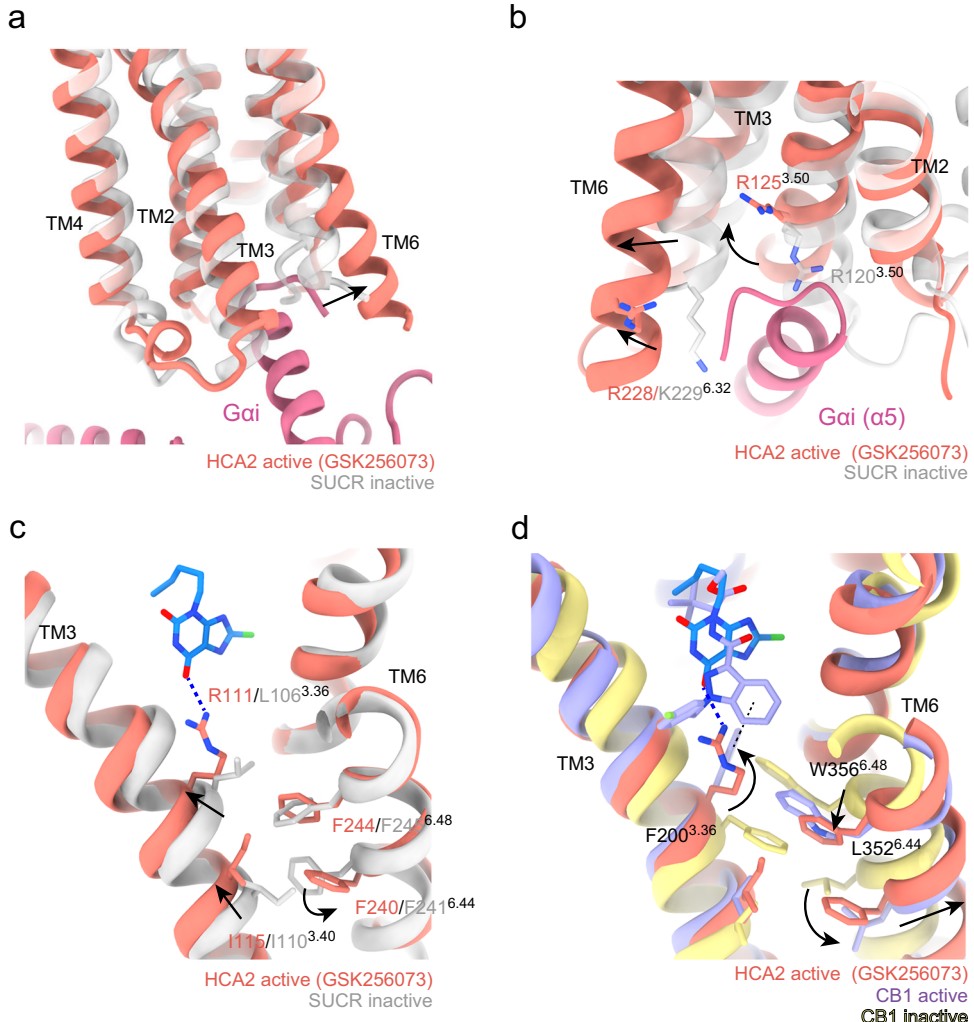

**Fig. 3 | Activation of HCA2. a** Superposition of the active conformation of HCA2 (red) with an inactive conformation of SUCR (gray, PDB: 6Z10). **b** Close-up view around the C-terminal helix of Gαi. Black straight arrows indicate outward movement of TM6. Black curved arrows indicate a positional shift of conserved side chains. **c** The interaction of R111[3.36] with GSK256073 and the difference in the PIF motif. **d** Superposition of the HCA2-activation switch with classical "twin toggle switches" of active CB1R (purple, PDB: 6N4B) and inactive CB1R (yellow, PDB: 5TGZ).

inactivated SUCR structure (Fig. 3b). Conformational rearrangement of R[3.50], known as an important residue in the DRY motif, is triggered simultaneously and has been confirmed in all GPCR-G protein complex structures[34–36].

Compared with the inactive SUCR structure, the interaction of R111[3.36] with the ligand causes TM3 to shift upward toward the extracellular side (Fig. 3c). The interaction between R111[3.36] and the agonist may induce a rearrangement of the side chain orientation of I115[3.40] and F240[6.44] (PIF motif)[15]. The F244[6.48] (known as a toggle switch, CW(F)xP motif) rotamer is consistent with F245[6.48] of inactivated SUCR. In contrast, most class A GPCRs have hydrophobic residues at position 3.36. A typical example is CB1, which has F200[3.36] and forms a hydrophobic interaction with W356[6.48], and agonist binding causes the side chains to flip relative to each other (Fig. 3d)[23,37]. Indeed, the arrangement of the residues on the active-state CB1 is consistent with the structure of HCA2-Gi (Fig. 3d).

HCA2 and SUCR are classified in the δ branch of class A GPCRs, including P2Y1[38], P2Y12[39], CysLT1/2[40,41], PAR1/2[42,43], PAFR[44], GPR35[30], and LPA6[45]. We investigated whether the conformation changes observed in comparison with SUCR are conserved features among class A GPCR family members by comparing the structures of the reported δ-branch GPCRs. Most class A GPCRs have Trp residues at position 6.48 in TM6, which recognize their ligands and initiate the

conformational changes required for receptor activation, while the δ-branch receptors have Phe or Tyr residues instead of Trp at this position (Supplementary Fig. 12a). Focused on the conformational change near the microswitch upon agonist binding, the upper half of TM5 of HCA2 bends inward starting at conserved Pro[5.50], and upward shifts of TM3 are commonly observed in all the known structures of the δ-branch GPCRs (Supplementary Fig. 12b-i). The rotamer of F[6.44] and F[6.48] adopts a "downward" conformation in all the structures, and TM6, including these residues, in the active HCA2 is pushed outward. The conformation observed in HCA2 is also consistent with that of G13-bound GPR35 (Supplementary Fig. 12j), suggesting that other δ-branch receptors have similar activation mechanisms.

## HCA2 ligand entrance
The structure of HCA2 shows that ECL2 on the extracellular side has a conserved β-hairpin structure that passes over the ligand-binding pocket and connects to TM5 (Fig. 4a). C177[45.50] of ECL2 forms a disulfide bond with C100[3.25] of TM3, which is conserved in many class A GPCRs and is essential for the formation of the ligand binding pocket[45,46]. The N-terminus of HCA2 also forms a β-hairpin structure, in which C18 and C19 form disulfide bonds with C266[7.25] and C183[5.33], respectively (Fig. 4a). Mutations on C100[3.25] and C177[45.50] cause a loss of activity and significantly reduce receptor surface expression levels,

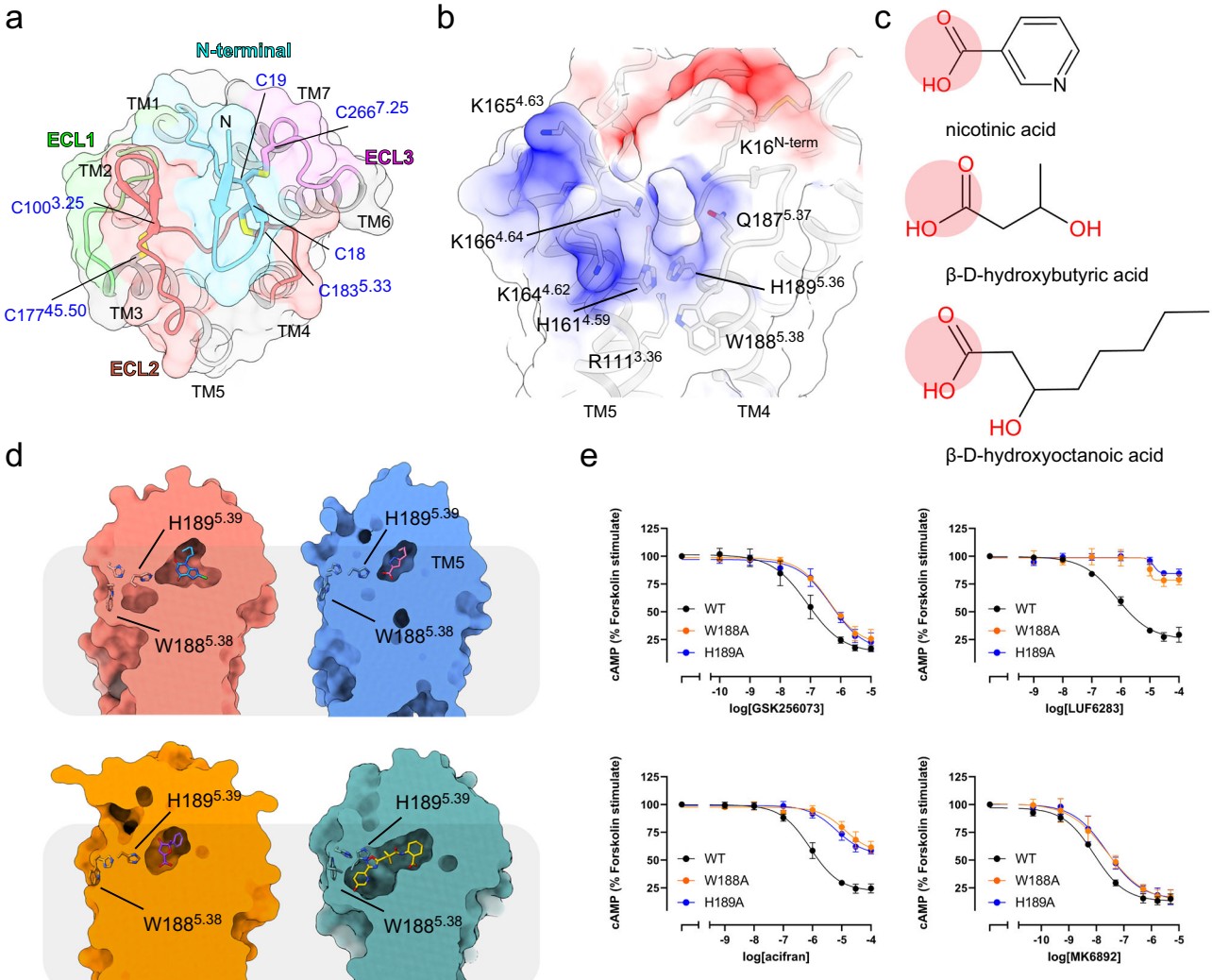

**Fig. 4 | Ligand entrance of HCA2. a** Top view of the agonist-bound HCA2 in cartoon representation and transparent surface representation. The extracellular loops are colored individually (N-terminus, cyan; ECL1, green; ECL2, orange; ECL3, pink) Six cysteine residues forming disulfide bonds are shown in stick representation. **b** Electrostatic potential surface of HCA2 at the extracellular side, ranging from -10 $kT/e$ (red) to +10 $kT/e$ (blue). Residues around the ligand entrance are labeled and shown in stick representation. **c** Two-dimensional representation of chemical structures of nicotinic acid, β-hydroxybutyrate, and β-hydroxyoctanoic acid. **d** Structural comparison in the cross-sectional views of the cryo-EM densities of HCA2 bound with GSK256073 (upper left), LUF6283 (upper right), acifran (under left), and MK6892 (under right). Models of the agonist are shown in stick representation, with their densities omitted to show the ligand binding pockets more clearly. **e** Concentration-response curves of cAMP inhibition assay for W188A and H189A mutants. Data are mean ± SEM of 3 independent experiments ($n = 3$). Source data are provided as a Source Data file.

suggesting that the 2 cysteines are essential for folding and trafficking of the receptor[19]. Mutations on C266[7.25] and C183[5.33], significantly reduce receptor activity but do not affect membrane localization[19], suggesting a crucial role for the additional disulfide bonds in proper ligand pocket construction.

On the basis of our structure, the agonist of HCA2 is completely occluded in the ligand pocket (Fig. 4a). It is structurally unfavorable for both polar endogenous ligands and synthetic agonists to enter the orthosteric binding site 'from the top' as non-lipid ligands normally do in GPCRs. Investigation of the charge properties in the extracellular region of HCA2 revealed the presence of a highly basic region located at TM4 and TM5 (Fig. 4b). Nicotinic acid, β-hydroxybutyrate, and β-hydroxyoctanoic acid are negatively charged in the endogenous environment (Fig. 4c). Considering the extended ligand-binding pocket in the MK6892-bound structure, we hypothesize that the gap between TM4 and TM5 and ECL2 is the entrance to the ligand pocket. Therefore, we focused on H188 and W189, which are located at the possible ligand entrance (Fig. 4d). We speculate that alanine mutants of those residues

are unable to completely close the ligand entrance and thus reduce the ligand response. In support of our hypothesis, W188A and H189A decreased the potency, even though these residues are distant from the binding site of GSK256073, LUF6283, and acifran (Fig. 4e). To assess the functional role of the potential ligand entrance, we performed MD simulations of HCA2, with and without GSK256073. The distance between L158[4.56] and W189[5.38] was essentially the same as that of the initial structure for about 900 ns in the presence of GSK256073, whereas under ligand-free conditions, the distance increased (Supplementary Fig. 13a, b). This observation may indicate that the upper half of TM5 itself shows higher flexibility without ligand binding.

The finding that the alanine mutants for H189 and W188 did not affect the activity of MK6892 may be due to the broad interaction of MK6892 with HCA2 (Fig. 4e). These results suggest that the space between TM4 and TM5 and ECL2 is the ligand entrance in HCA2, and endogenous agonists and synthetic agonists insert through the flexible open/close entrance and are occluded in the orthosteric binding pocket.

## HCA2 and HCA3 ligand selectivity

HCA3 (also known as GPR109B) is a subtype identified in hominids but not in mice or rats[46]. The difference between the 2 receptors within the sequence of the core TM domains (from TM1 to TM7) is only 15 amino acids, most of which are concentrated in the extracellular region (Fig. 5a). Another difference is that the C-terminus of HCA3 is 24 amino acids longer (Supplementary Fig. 14).

First, in the structure of the HCA3-Gi complex, we identified the difference in the acifran binding position (Fig. 5b). Similar to the acifran-bound HCA2 structure, Y284 and R111 form a polar interaction with the carboxylic group of acifran. In addition, the aromatic ring of acifran is surrounded by a hydrophobic pocket formed by V83[2.60], Y86[2.64], W93[ECL1], L104[3.29], F107[3.32], and I178[ECL2] (Fig. 5b, c). Indeed, mutations of Y284A, R111A, and W93A retained their expression levels, but completely abolished receptor activity, and Y86A and F107A reduced the potency and efficacy (Fig. 5d and Supplementary Fig. 9b). On the other hand, V103A increased the potency.

We then compared the binding pose of acifran in HCA2 and HCA3 and found that while the binding positions of acifran were almost identical, the binding poses were slightly different (Fig. 5c). We focused on the surrounding amino acids to elucidate what might be responsible for these differences in acifran binding poses (Fig. 5c). The amino acids with the most notable difference in the ligand binding pocket of HCA3 are V83[2.60] (L83 in HCA2), Y86[2.64] (N86), S91[ECL1] (W91), and V103[3.28] (M103). The aromatic ring of acifran forms a hydrophobic interaction with W91 in HCA2, and Y86 plays a similar role in HCA3. Based on our structures, the ligand-binding pocket can be roughly classified into 3 sub-pockets (pockets I-III, Fig. 5e, f). The side chains of L83 and M103 in HCA2 are larger than those of V83 and V103 in HCA3, and the size of pocket I in HCA2 is narrower, leading acifran to fit deeply into pocket III and interact stably with R111 and Y284 (Fig. 5e). On the other hand, in HCA3, pocket I cannot stably hold the aromatic ring of acifran due to the larger volume (Fig. 5f), consistent with a 10-fold decrease in affinity[2]. The structures of HCA2 and HCA3 can also explain the selectivity of GSK256073. Replacement of W91[ECL1] in HCA2 with Ser shifts F277[7.36] closer to the ligand-binding pocket in HCA3 than in HCA2, and the distance between Y87[2.64] and Y284[7.43] in pocket II is slightly closer (Fig. 5f). These results show that the narrowing of pocket II causes steric clashes in the purine-2,6-dione part of GSK256073, supporting the previous report that the affinity is 100-fold lower for HCA3[17] (Fig. 5f). Together, among the 15 different amino acids, the 6 residues at positions 83, 86, 91, 103, 107, and 178 contribute to the agonist selectivity of HCA2 and HCA3.

## Gi coupling interface

While HCA2 is reported to couple with only Gi, no clear mechanism for this selectivity has been reported. Because the Gi-binding interfaces of the 4 HCA2-Gi structures are almost identical, the GSK256073-bound structure with the highest resolution was used to examine the selectivity in the following section. Our structures indicate that the binding of HCA2 to Gi uses 4 interfaces (Fig. 6a). The first is between the cytoplasmic ends of TM3, TM5, and TM6 in HCA2, and the C-terminal α5 helix of Gi. V129[3.54] of TM3, I211[5.61] and I215[5.65] of TM5, and I233[6.37] and I226[6.30] of TM6 in HCA2 form hydrophobic interactions with L344, L348, and L353 of the Gi subunit (Fig. 6b). The hydrophobic interaction is a common feature of other Gi-bound GPCRs, including dopamine receptor 3[8] and cannabinoid receptor type 1[4]. The second is between intracellular loop 2 (ICL2) and the hydrophobic cleft composed of Gi (Fig. 6c), which is also observed in Gs- and Gi-coupled GPCRs. In general, hydrophobic residues are conserved at position 34.51. In HCA1-3, a histidine residue is located at this position and interacts with αN, the β2-β3 loop, and α5 in Gi to fit into the hydrophobic groove formed by L194, F336, and I343. The third is the interaction between ICL3 of HCA2 and the G protein. ICL3 of HCA2 is very short, but residues R218[ICL2] and R222[ICL2] in the receptor form a polar interaction with D337/D341 and

E318 in Gi, respectively (Fig. 6d). Fourth, K57 of ICL1 interacts with D350 (Fig. 6e). We introduced mutations into these amino acids and measured the Gi activity (Fig. 6g). Only R218A exhibited remarkably decreased Gi activity. The corresponding residues in other Gi-coupled GPCRs form polar interactions with D341 in Gi, suggesting a contribution to G protein selectivity[7,8,20]. R128[3.47], R222[ICL3], and H223[ICL3] are not critical for specific interactions with Gi. These residues may be required, however, for the positively charged property of these parts of the cytoplasmic side, which is a key driving force for coupling GPCRs and Gi proteins (Fig. 6f)[8,21].

## Discussion

HCAs are expressed on adipocytes and various immune cells and have recently attracted attention as potential drug targets for neurodegenerative diseases. In this study, we determined 5 agonist-bound structures of HCA2 and HCA3 in complex with Gi proteins using single-particle cryo-EM analysis. Combining these structures with biochemical analyses, we determined the possible ligand entry pathway, recognition, and activation mechanisms of HCAs. We found that the polar interaction of R111[3.36] located in the center of the ligand-binding pocket and the carboxyl of the ligand are essential for receptor activation. Further, the structures of HCA2 and HCA3 revealed ligand preferences based on differences in amino acids in the ligand-binding pocket. The extracellular side of HCA2 and HCA3 was completely covered by a lid-like structure and mutation and MD simulation analyses suggest that the endogenous and synthetic agonist access from the gap between TM4 and TM5 and ECL2. These results provide structural pharmacologic insights into HCAs.

To gain further insight into the mechanisms underlying the receptor activation, we compared our active-state structures with the recently published inactive HCA2 structure. First, a significant difference is that the TM5 helix is positioned more outward in the inactive structure relative to the cryo-EM structure (Supplementary Fig. 15a). Second, R111[3.36] and F180[ECL2], which are important for ligand binding, undergo major conformational changes. Furthermore, the microswitches (CW(F)xP and PIF motif) show similar changes as in the comparison of δ-branch class A GPCRs. (Supplementary Fig. 15c). Finally, a comparison of the binding interfaces to the C-terminal helix of the Gi subunit revealed a rearrangement of R125[3.50] (the DRY motif) and Y294[7.53] (the NPxxY motif) (Supplementary Fig. 15d). We found that F232[6.36] of TM6 is rotated about 80° relative to the inactive structure. Comparison with the inactive form of HCA2 is comprehensively consistent with our results on the activation mechanism.

Whether or not the metabolites bind from the lateral side of the receptors within membranes has not been fully established. Similar to our study, access to the orthosteric pocket between TM4 and TM5 has been proposed for 3 other δ-branch lipid receptors, PAFR, CysLT1, 2, and LPAR6. In PAFR, the ligand is jammed in the TM4-TM5 gate, leaving the ligand entrance open[44] (Supplementary Fig. 16a). The melatonin MT1 receptor also binds a hydrophobic ligand, which is proposed to enter through a gap between TM4 and TM5. In particular, the conformational change of Y187[5.38] in TM5 of MT1 during its transition to the activated state is suggested to reduce the size of the ligand entrance and ligand dissociation rate[47,48] (Supplementary Fig. 16b). In the HCA2 structures, W188[5.38] in TM5 is located in the groove between TM4 and TM5, completely closing off ligand access (Fig. 6a, b and Supplementary Fig. 16c). In addition, MD simulations suggest that the TM5 helix is more flexible in the absence of a ligand. These residues may be involved in the ligand dissociation rate of HCA2.

HCA2 and HCA3 have different ligand selectivities despite having >90% sequence identity. The difference in the amino acid sequence between the 2 receptors is 15 residues except for the C-terminus. We showed that 6 amino acids (positions 83, 86, 91, 103, 107, 178) contribute primarily to the volume and shape of the ligand-binding pockets. The volume of sub-pocket I of HCA2 is smaller than that of

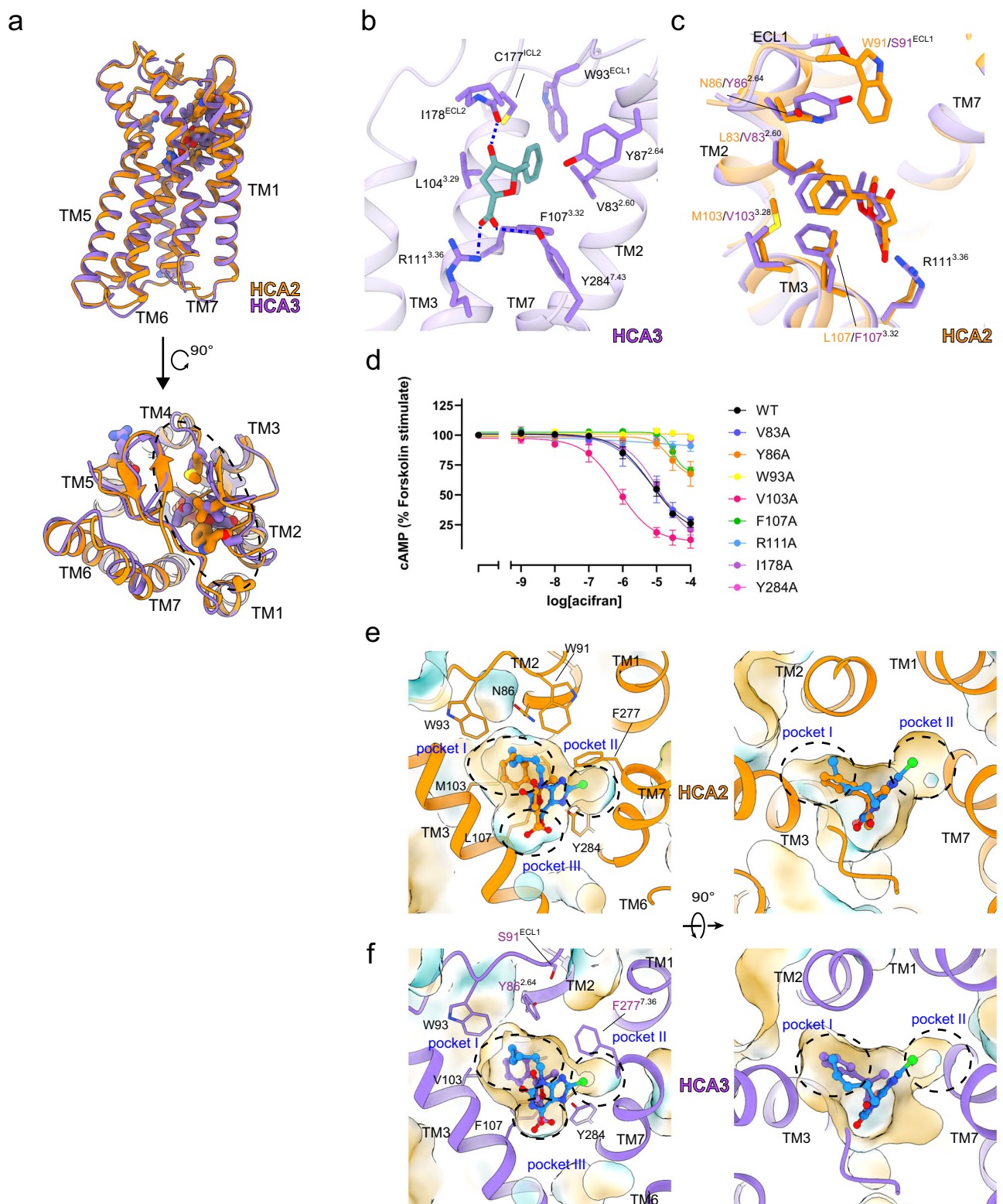

**Fig. 5 | Structural comparison of HCA2 and HCA3. a** Overall structural comparison of active HCA2 and HCA3 in complex with acifran. Different residues are represented as stick models. **b** Detailed interactions of acifran in HCA3. Residues within 4 Å of acifran (green) are shown as stick models. Polar interactions are indicated by blue dashed lines. **c** Comparison of acifran binding poses in HCA2 and HCA3. Acifran bound to HCA2 and HCA3 is colored orange and purple, respectively.

**d** Concentration-response curves of cAMP inhibition assay for HCA2 mutants. Data are mean ± SEM of 3 independent experiments (*n* = 3). Source data are provided as a Source Data file. **e**, **f** Surface representation of ligand-binding pocket of HCA2 (**e**) and HCA3 (**f**). Ligands are shown in stick representation, and acifran in each model is colored as in (**c**). GSK256073 is colored cyan as in Fig. 2c, d.

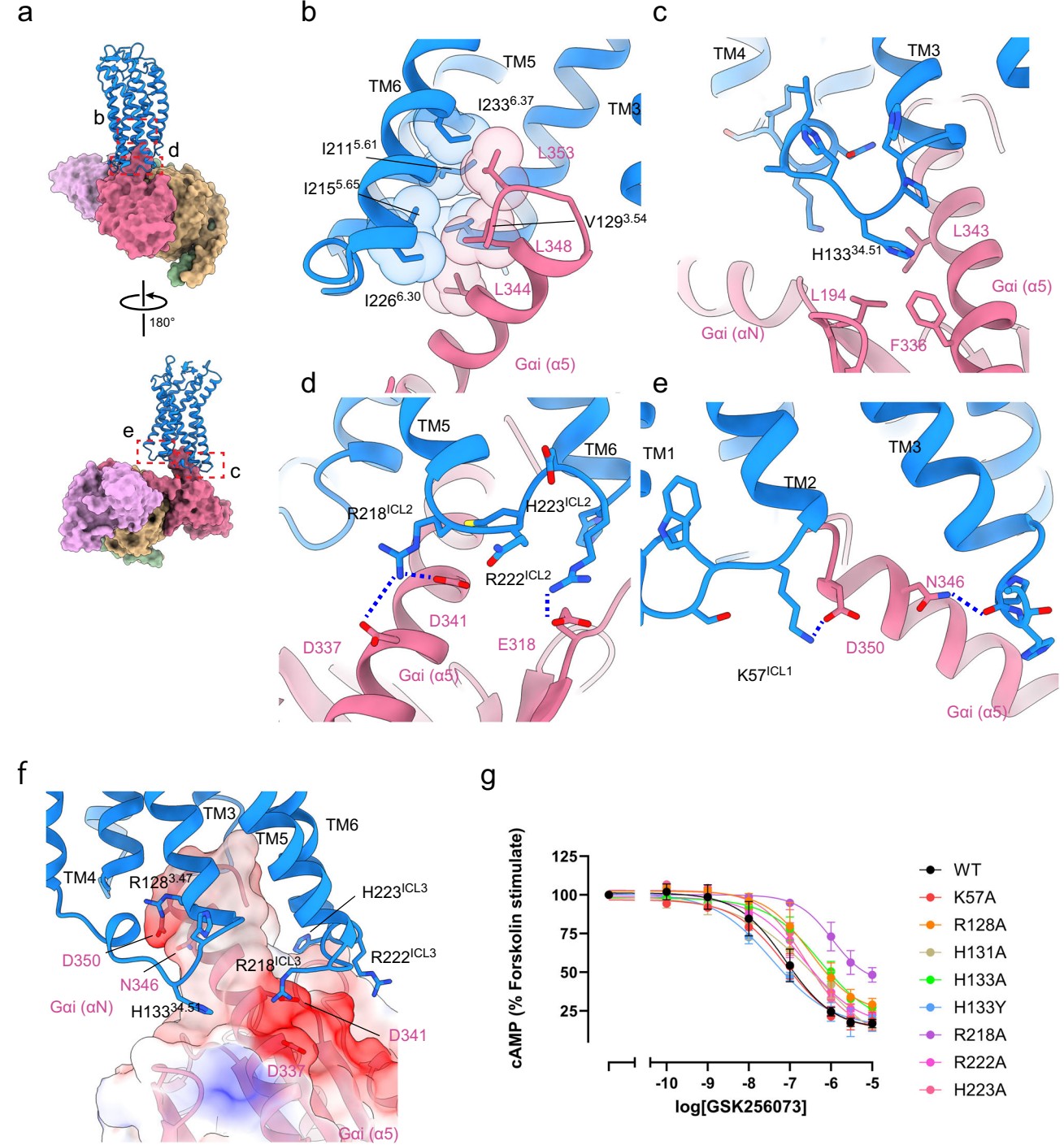

**Fig. 6 | The G protein interfaces of HCA2. a** Interface between the active form of HCA2 (cartoon representation) and the bound Gi protein (surface representation). **b** Hydrophobic interactions between HCA2 and the C terminus of the Gi subunit. **c** Interactions between ICL2 and a hydrophobic cleft formed by Gi. **d** Interactions of ICL3 with Gi. **e** Interactions of ICL1 with the C-terminus of Gi. Hydrophilic interactions are indicated by blue dashed lines. **f** Close-up view of the interface between HCA2 and Gi. Side chains of positively and negatively charged amino acids are shown in stick representation. The electrostatic potential surface of Gi ranges from −10 *kT/e* (red) to +10 *kT/e* (blue). **g** cAMP inhibition assay of Gi protein-binding interface mutants. Data are mean ± SEM of 3 independent experiments (*n* = 3). Source data are provided as a Source Data file.

HCA3, favoring smaller molecules such as niacin and β-hydroxybutyrate, whereas 3-hydroxyoctanoic acid, an endogenous agonist of HCA3, has a longer carbon chain than β-hydroxybutyrate, an endogenous HCA2 agonist. The N86Y/M103V/L107F triple mutant of HCA2 is reported to increase its activity by β-hydroxyoctanoate to levels comparable to those of HCA3[3], which can be explained by the extended sub-pocket I. HCA3 is also activated by lactic acid bacteria-derived metabolites such as D-PLA and ILA[49], but these amino acids do not activate HCA2. These compounds are slightly larger and more hydrophobic, which may favor HCA3. The MK6892-bound structure, however, shows that the ligand-binding sites in GPCRs are highly flexible. More structures of HCAs in complex with various ligands will provide clues to elucidate the ligand selectivity and potentially drive more efficient structure-based drug design.

A side effect of drugs targeting HCA2 is a symptom called the niacin flush, which is reported to be a β-arrestin1-mediated effect[50]. In recent years, bias agonists that activate only specific downstream signals have attracted considerable attention and are expected to lead to the development of drugs with fewer side effects[51–53]. A similar strategy could be adapted for HCA. In the future, structural and biophysical analyses of the HCA2-βarr1/2 complex will be beneficial to gain insight into bias signaling based on our results. This study provides a structural framework for understanding the signaling systems of the HCA family and is expected to facilitate structure-based subtype selective drug discovery targeting HCAs, such as HCA2 and HCA3.

## Methods

### Constructs
Human HCA2 (residues 1–363) and HCA3 (residues 1–387) were cloned into a pFastBac1 vector with an N-terminal hemagglutinin (HA) signal peptide and a FLAG-tag, followed by thermostabilized apocytochrome b562RIL (bRIL). For HCA2, the LgBiT subunit was fused to the C terminus to stabilize the complex. Gαi1 (Gi) with 2 dominant-negative mutations (G203A, A326S), Gβ1, Gβ-SmBiT, Gγ2, and a single-chain antibody scFv16 (a kind gift from Dr. Brian Kobilka at Stanford University) were individually cloned into pFastBac1 vectors. To generate mutants for cAMP measurement, the site directed mutation was introduced in HCA2 and HCA3 using primers shown in Supplementary Table 2.

### Protein expression and purification
HCA2 and HCA3, Gi, Gβ1, Gγ2, and scFv16 were co-expressed in Sf9 insect cells using the Bac-to-Bac Baculovirus Expression System (Thermo Fisher Scientific). Cell cultures were grown in Sf900-II SFM medium (Thermo Fisher Scientific) to a density of $2–3 \times 10^6$ cell/mL and then infected with the viruses expressing HCA2 or HCA3, Gαi1, Gβ1, Gγ2 and scFv16. Cell culture was collected by centrifugation 48 h post-infection and stored at −80 °C.

Cell pellets were lysed by homogenization in 20 mM HEPES (pH 7.4), 50 mM NaCl, 10 mM MgCl2, 10% glycerol, 25 mU/mL apyrase (New England Biolabs), and Protease Inhibitor Cocktail (Roche). After incubation at room temperature for 1 h, the membranes were solubilized by the addition of 0.5% (w/v) lauryl maltose neopentylglycol (LMNG, Anatrace) and 0.03% (w/v) cholesteryl hemisuccinate Tris salt (CHS, Anatrace) for 2 h at 4 °C. The supernatant was cleared by centrifugation and incubated with M2 FLAG resin (Sigma-Aldrich) for 1 h. After binding, the resin was washed with 10 column volumes of 20 mM HEPES (pH 7.4), 100 mM NaCl, 5 mM MgCl2, 0.05% LMNG, 0.05% glycodiosgenin (GDN, Anatrace), and 0.003% (w/v) CHS. The complex was then washed with 10 column volumes of 20 mM HEPES (pH 7.4), 100 mM NaCl, 5 mM MgCl2, 0.01% LMNG, 0.01% GDN, 0.0006% (w/v) CHS. The complex was eluted in the second wash buffer containing 200 μg/mL FLAG peptide (Sigma-Aldrich). The eluted complex was supplemented with 100 μM TCEP (Fujifilm Wako) for reducing conditions. The complex was purified by size-exclusion chromatography on a Superose 6 10/300 column (Cytiva) in 20 mM HEPES (pH 7.5), 100 mM NaCl, 0.00075% LMNG, and 0.00025% GDN with 0.00004% CHS and 100 μM TCEP. Peak fractions were concentrated to ~15 mg/mL for cryo-EM. We added each ligand (GSK256073, MK6892, LUF6283, acifran) for stable complex formation throughout the purification procedure.

### Cryo-EM grid preparation and data collection
Purified HCA2-Gi and HCA3-Gi complex (3.5 μL) was applied to glow-discharged 300 mesh R1.2/1.3 grid (Quantifoil). Support films were plunge-frozen in liquid ethane using a Vitrobot Mark IV (Thermo Fisher Scientific) with a 10-s hold period, blot force varying between 0 - 10, and blotting time of 4 s while maintaining 100% humidity and 4 °C. For HCA2 with LUF6283 and acifran, and HCA3 with acifran, cryo-EM data were collected at 300 kV using a JEM-Z320FHC electron microscope (JEOL) equipped with a K2 Summit direct electron detector in the electron counting mode using SerialEM[54]. The calibrated pixel size was 0.99 Å (for HCA2) and 0.78 Å (for HCA3) on the specimen level, and exposures of 8 s were dose-fractionated into 40 frames with an electron flux of 8 e-/pix/s (for HCA2) or 5 e-/pix/s (for HCA3). For HCA2 with GSK256073 and MK6892, cryo-EM data were collected using a Titan Krios Gi3 (Thermo Fisher Scientific) with a BioQuantum energy filter (Gatan) and a K3 direct electron detector (Gatan), operated using EPU software (Thermo Fisher Scientific) running a 3 × 3 image shift pattern at 0° stage tilt. A nominal magnification of 105,000 x was used in CDS mode with a calibrated pixel size of 0.83 Å, and exposures of 4.5 s were dose-fractionated into 49 frames with an electron flux of 7.55 e-/pix/s.

### Cryo-EM data processing
Datasets for HCA2-Gi and HCA3-Gi complexes were processed using CryoSPARC[22] and Relion[55]. For GSK-HCA2-Gi, MK-HCA2-Gi, LUF-HCA2-Gi, acifran-HCA2-Gi, acifran-HCA3-Gi complexes, a total of 4872, 5802, 2828, 3602, and 6723 image stacks, respectively, were subjected to beam-induced motion correction using MotionCor2.1[55] in Relion-4.0. Contrast transfer function parameters for each micrograph were estimated by the CryoSPARC patch CTF algorithm. Particles were autopicked using reference-based picking, extracted with a box size of 288 pixels, and subjected to several rounds of 2D classification to remove contaminants. Initial maps were generated using stochastic gradient descent-based multi-ab initio reconstruction in CryoSPARC. A good class was subjected to 3D non-uniform refinement[56] and particles were re-imported to RELION-4.0. Particles were then subjected to 3D refinement in RELION-4.0, followed by CTF refinement and Bayesian polishing. After successive 3D refinement, we performed focused 3D classification with a mask only including the TM domain without alignment. Particles of the best class were then imported back to CryoSPARC for 3D non-uniform refinement and 3D local refinement. The final dataset contained 245,517, 181,958, 181,273, 146,577, and 95,567 particles, which generated maps with resolutions at 2.85, 2.97, 3.13, 3.17, and 3.18 Å, respectively. The resolution of these maps was estimated internally in CryoSPARC by gold-standard Fourier shell correlation using the 0.143 criterion. Local resolution estimation was performed with the CryoSPARC local resolution estimation algorithms using half maps.

### Model building and refinement
Model building and refinement were carried out using an Alphafold2[57] predicted structure as a starting model, which was fitted into the HCA2-Gis map using UCSF ChimeraX[58]. A draft model was refined by iterations of real space refinement in Phenix[59] and manual refinement in Coot[60]. The ligand model was generated with the Grade web server (https://grade.globalphasing.org), docked using Coot, and refined in Phenix. Final map model validations were carried out using Molprobity in Phenix.

### Bioluminescence resonance energy transfer assays (BRET2)
BRET2 assays were performed and analyzed as previously described with the following modifications[61]. Expi293 cells grown in pro293s suspension media (LONZA) were transfected at a density of 1 million cells/ml in a 2-ml volume using 1200 ng total DNA at a 1:1:1:1 ratio of receptor/Gα-rLuc8/Gβ/Gγ-GFP2 and a DNA/polyethyleneimine ratio of 1:5, and incubated in a 6-well plate at 220 r.p.m., 37 °C for 48 h. Cells were harvested by centrifugation, washed with Hank's balanced salt solution (HBSS) (Thermo Fisher Scientific), and resuspended in assay buffer (HBSS with 20 mM HEPES pH 7.5 and 0.01% bovine serum albumin [BSA] (Fujifilm Wako)) with 5 μg/ml of freshly prepared coelenterazine 400a

(biotium). Cells were then placed in white-walled, white-bottom 96-well plates (Corning) in a volume of 60 µl/well. Drug dilutions were prepared in assay buffer, and 30 µl was immediately added to plated cells. Plates were read using a SpectraMax i3x multi-plate reader (Molecular Devices) using 410- and 515-nm emission filters with a 1-s integration time per well. The computed BRET ratios (GFP2/RLuc8 emission) were normalized to a ligand-free control (net BRET). Data were analyzed using "Sigmodal, 4PL, X is concentration" in GraphPad Prism 10.0.

### cAMP inhibition assay

The inhibitory effects of different HCA2 and HCA3 constructs or mutants on forskolin-induced cAMP accumulation were measured using the GloSensor cAMP assay (Promega). Expi293 cells were transiently co-transfected with the GloSensor and various mutants of HCA2 plasmids using PEI in 6-well plates. After incubation at 37 °C for 24 h, the transfected cells were harvested and washed with Dulbecco's phosphate-buffered saline (PBS), centrifuged at 190×g for 5 min, and suspended in HBSS (Thermo Fisher Scientific) containing 0.01% BSA and 5 mM HEPES (pH 7.4). The cells were then resuspended in 2% GloSensor cAMP reagent (Promega) at room temperature for 2 h. The cell suspension was seeded into a 96-well white plate (Corning) at a volume of 80 µL per well. A 10-µL volume of 10 x drug buffer diluted in HBSS containing HEPES and 0.01% BSA was added to each well; the plates were incubated for 10 min. Then, a 10-µL volume of 10-uM forskolin (Sigma-Aldrich) (final 1 µM) was added and the plates were incubated for 20 min at room temperature. The luminescence of the cells was measured using a SpectraMax i3x multi-plate reader. Data were analyzed using "Sigmodal, 4PL, X is concentration" in GraphPad Prism 10.0.

### Surface expression

Cell-surface expression of HCA2, HCA3, and its mutants were measured by ELISA chemiluminescence. In brief, 48-h post-transfected cells plated in 96 white-well plates were fixed with 50 µl per well 10% (v/v) formaldehyde for 10 min at room temperature. The cells were then washed twice with 80 µl per well of PBS and incubated with 50 µl per well 5% (v/v) BSA in PBS for 1 h. Cells were incubated with an anti-Flag–horseradish peroxidase-conjugated antibody (Sigma-Aldrich, A8592) diluted 1:10,000 in 5% (v/v) BSA in PBS for 1 h at room temperature. After washing 3 times with 80 µl per well PBS, 50 µl per well Super Signal Enzyme-Linked Immunosorbent Assay Pico Substrate (Thermo Fisher Scientific, 37070) was added to each well for development of the signal and the luminescence was counted using a SpectraMax i3x multi-plate reader. The luminescence signal was analyzed in GraphPad Prism 10.0 and data were normalized to the signal of wild-type HCA2 or wild-type HCA3.

### MD simulation

Input models and parameters for all-atom MD simulations of HCA2 with and without GSK256073 were prepared using CHARMM-GUI[62,63] and CHARMM-GUI Membrane Builder[64]. All the molecules except HCA2 and GSK256073 were removed from the cryo-EM models. The N-terminus and C-terminus were capped by acetylation and methylamidation, respectively. Protonation states of the titratable residues at pH 7.0 were determined with PROPKA[65]. All the aspartic acid and glutamic acid residues were negatively charged except D73, D290, E37, and E196. All the lysine and arginine residues were positively charged. All the histidine residues were neutralized by δ-nitrogen protonation except H189, which was protonated at the ε-nitrogen. The model was embedded in a rectangular lipid bilayer comprising 259 POPC molecules with the orientation estimated by PPM2.0[66]. The system was solvated and charge-neutralized by ~22,000 water molecules and 150 mM NaCl. The system dimension was 100 Å × 100 Å × 114 Å and the total number of

atoms was ~106,000. The CHARMM36m force-field parameters[63] were used for the proteins, lipids, and ions. The TIP3P model[67] was used for water. An initial topology and force-field parameter for the GSK256073 was generated using the CHARMM general force field (CGenFF)[68] and CHARMM-GUI Ligand Reader & Modeler[69]. The atomic partial charges and geometric parameters were optimized and validated with FFParam[70] using Gaussian16[71] and CHARMM[72] as QM-MM backends.

MD simulations were performed using GROMACS 2022.5[73]. First, the system was energy-minimized according to the protocol generated by CHARMM-GUI. Then, 3 independent MD runs were performed. For each run, a 900-ns constant-NPT production run was performed after 6 steps of equilibration runs according to the protocol generated by CHARMM-GUI. During the production run, the temperature and pressure were kept at 310.15 K and 1 bar, respectively, using the Nosé-Hoover thermostat[74,75] and the Parrinello–Rahman barostat[76]. Bond lengths involving hydrogen atoms were constrained using the LINCS algorithm[77,78]. Long-range electrostatic interactions were calculated with the particle mesh Ewald method[79,80]. The simulations were carried out using the supercomputer "Flow" at Information Technology Center, Nagoya University. The trajectory analysis was performed using the MDAnalysis library[81].

### Structure and sequence comparisons

Sequence alignment was performed using the GPCRdb (https://gpcrdb.org/) and the representation of the sequence alignment was generated using the ESPript website (http://espript.ibcp.fr)[82]. The generic residue numbering of GPCR is based on the GPCRdb (https://gpcrdb.org/).

### Statistics and reproducibility

All functional study data were analyzed using GraphPad Prism (GraphPad Software) and are presented as means ± SEM from at least $n = 3$ independent experiments performed.

### Reporting summary

Further information on research design is available in the Nature Portfolio Reporting Summary linked to this article.

## Data availability

All data generated or analyzed in this study are included in this article and Supplementary Information. The cryo-EM density maps and corresponding coordinates have been deposited in the Electron Microscopy Data Bank (EMDB) and the PDB, respectively, under the following accession codes: PDB-8IHB and EMDB-35442 (GSK256073), PDB-8IHF and EMDB-35443 (MK6892), PDB-8IHH and EMDB-35444 (LUF6283), PDB-8IHI and EMDB-35445 (Acifran), and PDB-8IHJ and EMDB-35446 (HCA3-Gi with Acifran), and PDB-8IHK and EMDB-35447 (HCA3-Gi with Acifran (local)). Source data are provided with this paper.

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

## Acknowledgements

This work was supported by Grants-in-Aid for Scientific Research (A) under Grant Number 20H00451 the Japan Agency for Medical Research and Development (AMED) under Grant Number JP21ae0121028 (Y.F). This work is partially supported by Platform Project for Supporting Drug Discovery and Life Science Research (Basis for Supporting Innovative Drug Discovery and Life Science Research [BINDS]) from the Japan Agency for Medical Research and Development (AMED) under Grant number JP21ama0101074. Grants-in-Aid for Scientific Research (B) and a Grant-in-Aid for Challenging Exploratory Research (A.O.).

## Author contributions

S.S. and Y.F. designed the research. SS designed the expression constructs, purified the HCA2-Gi and HCA3-Gi complexes, and performed the cAMP, BRET2 assay and cell surface expression of the HCA2 and HCA3 mutants. S.S., K.N. and H.S. collected the cryo-EM data using an automated data acquisition system. S.S. processed the cryo-EM data and built and refined the structure models. K.T. performed the MD simulations. S.S., H.S., A.O. and Y.F. wrote the manuscript. Y.F. supervised the research. All authors read, revised, and approved the manuscript.

## Competing interests

The authors declare no competing interests.
