## [Peer Review File · Nature Communications]

Structural basis of hydroxycarboxylic acid receptor signaling mechanisms through ligand bindingREVIEWER COMMENTS

Reviewer #1 (Remarks to the Author):

The paper by Suzuki et al entitled “Structural basis of signaling mechanisms of 2 hydroxycarboxylic acid receptors through ligand binding” describes the ligand binding mechanism of the HCA2 and HCA3 receptors with multiple agonists. The authors performed cryo-EM and tested their structural findings using mutagenesis and MD simulations. This is the first study describing the structures of the HCA receptors and it provides a valuable framework for understanding ligand binding and activation. While this is a really good study with gorgeous cryo-EM maps at pretty high resolution that are supported by extensive mutagenesis experiments, I have significant concerns about the quality of the models. While in itself it does not affect the conclusions of the paper, they are quite serious and need to be addressed before the publication and before the coordinates are available to the general public.

The major issues with the PDB models are outlined below.

1. Rotamers:

- a. For all models there is an extremely low number of of favored rotamers (e.g. ~70% for 8IHK, 65% for 8IHB (and others too) when this should be closed to 98%).
- b. Number of rotamer outliers is also too high for 8IHB and 8IHF considering the high quality of the maps

2. Clashscores

- a. Very high clash scores for 8IHH, 8IHI, 8IHJ and 8IHK.
- b. Numbers in Table 1 do not correlate to the Molprobit numbers – e.g. 12.45 for clashscore for 8IHK vs 8.92 in the table 1

3. Ramachandran

- a. Very suspicious distribution of -3.42 ± 0.43 for 8IHK (possibly others as I havent checked all the models) suggesting that the Ramachandran angles were forced into the “favoured” space without consulting the residue environment
- b. Favoured Ramachandran – too low for 8IHF and 8IHK
- c. Please switch out which are favoured, allowed and disallowed in Table 1 (this is minor – just a typo)

4. Ligands

- a. Wrong stereochemistry of GSK256073 – around the N atom – needs to be planar – then the lipid tail will actually fit

b. LUF6283 – doesn't fit to the density well

5. Why are coordinates different between the receptor in 8IHJ and 8IHK?? They are the same receptor and the same map (only local vs. global.)

Minor modelling issues:

6. MMK6892- check NAG stereochemistry/conformation (maybe check other models for the same issues as I haven't check all of them)

7. All structures – add a carboxylic acid to the end of Galpha

Comments about the manuscript itself:

1. While this is a structural biology paper and structures are the main point of this study, to me it felt that there is an overwhelming amount of structural discussion that is targeted to a very specialist audience (discussion about receptor activation, G protein coupling interfaces etc.). This amount of information and detail makes this for a very difficult read, and at times the point authors are making (e.g. about the selectivity and others) gets lost. Perhaps some of this discussion can be shortened and/or moved to a supplement?

2. Line 96-97 Unclear why authors compare HCA receptor binding site to that of the amine-coupled receptors.

3. Line 158 “the inactivated AlphaFold2 (AF2) structure” perhaps authors meant “inactive”?

4. Line 240 “The aromatic ring of acifran forms a hydrophobic interaction with W91 in HCA2” – unclear how is this happening as W91 is >4Å away ?

5. Line 253 “suggesting that the affinity of acifran is approximately 10 times higher for HCA2 than for HCA3” -> perhaps authors meant “resulting in”?

Reviewer #2 (Remarks to the Author):

In the present paper Suzuki and coworkers present four novel cryo-EM structures of the hydroxycarboxylic acid receptor 2 (HCA-2), in complex with heterotrimeric Gi and four different agonist molecules: 1) the 40-year old niacin analog Acifran (used for dyslipidemia), 2) the partial agonist LUF6283, 3) the high affinity, and highly selective GSK256073 (having been in phase-2 clinical trials for diabetes); and 4) the high affinity, highly selective, late stage Merck agonist MK6892. The authors also present the first cryo-EM structure of the closely related HCA-3 – only found in humanoids - in complex with heterotrimeric Gi and the dual specific Acifran agonist. The structural analysis is complemented by a comprehensive mutational and signal transduction analysis of a long list of residues involved in ligand

or G protein binding and receptor activation in both receptors. The active HCA2 cry-EM structures are compared to an X-ray structure of the inactive form of a related receptor, SUCNR1 and to an Alpha-fold generated model of inactive HCA2 itself.

This paper constitutes an amazing amount of solid structural biology work and information plus carefully performed molecular pharmacological work on two interesting/important receptors, of which HCA-2 – originally known as the niacin receptor - has been a major drug target in the pharmaceutical industry aiming at changing medical indications over the years: dyslipidemia, diabetes, neurological inflammation etc. and consequently a rich pharmacology to characterize as here done very elegantly.

Unfortunately, a recent publication also in Nature Comm. by Yang and coworkers appears to steal some of the thunder of this story. However, that is only from a superficial point of view as they only have HCA-2 in complex with MK6892 (basically identical to the present structure) plus a mutationally ‘frozen’ inactive HCA-2 structure and consequently are only able to tell a rather simple story of yet another ligand bound in a GPCR. Yang and coworkers are, due to their lack of the three other ligand-receptor complexes, unable to tell and address the interesting and novel story of how polar ligands in HCA receptors get access to a deep pocket, which is totally secluded from the extracellular aqueous phase. That is, a story which is addressed in the present version of this paper, but which result-wise has been hidden in the supplementary material and only is told in relatively few lines in the Results and Discussion.

Major Point:

1. The authors should simply restructure the paper to make their proposed novel ligand entry the major theme. The data presented in supplementary Fig 15 should be included in the main paper and presented and discussed in much more detail. You should focus on the story of the positively charged lateral entry path for these anionic, polar ligands from the lipid head group area as the main novel message of the paper. That is, instead of just another long presentation and discussion of ligand binding in an orthosteric binding pocket and a similar story of the receptor G protein interactions. Neither of these stories providing significant novel information – just as they did not in the Yang paper and just as they do not in most of the many cryo-EM GPCR papers which currently are being published. Your paper can be truly different by e.g. telling that: 1) It is structurally impossible for both the polar endogenous ligands (see point #2) as well as for the synthetic agonists to enter the orthosteric binding site ‘from the top’ as non-lipid ligand normally do in GPCRs; 2) All four HCA-2 agonist are firmly bound deeply below a totally ‘locked lid’ interacting with the crucial R3.36 (see minor point concerning deepness) ; 2) The ‘lid’ is composed of ECL-2 and the N-terminus of the receptor ‘knitted together’ by three disulfide bonds which already are mutationally addressed in your paper. 3) You have identified a lateral entrance channel between TM-IV and-V and demonstrated how it mouth and surrounding are electropositively charged 4) importantly, although e.g. GSK256073 occupies the central deep part of the orthosteric pocket it is nevertheless affected by mutations around the proposed lateral entry site, i.e. ‘far’ away from its final binding site. 5) Finally, the discussion (line 322-327) of the fact that receptor selectivity towards different types of ligands e.g. aromatic D amino acids surprisingly cannot be explained by differences in the orthosteric binding pocket, could very likely instead be explained by differences in the entry path as demonstrated convincingly by MD simulations in the catecholamine receptors. You should

therefore at least compare the structure of the proposed entry path of HCA2 with that of HCA-3 – they are likely different.

2. An in-depth MD analysis of ligand entry and movements of residues in and around the entry channel would be really great to see. But it would be a huge study which will be more suited as a follow up study. The current MD simulations or rather the displayed data (Fig. 6D) concerning the movements of L258 relative to W188 with and without the GSL ligand in place are not very convincing. It is possible that other measures such as rotation of W188, which potentially could function as a ‘swing door’ in the entry channel would give more interesting and significant differences?

3. The whole background narrative needs to be seriously changed. 1) HCA receptors are not ‘short chain fatty acid receptors’. Their endogenous agonists are as their name tells, hydroxycarboxylic acids (lactate, β -hydroxybutyrate/ketone bodies, and β -hydroxyoctanoate). Some classical short chain fatty acids may also cross react with them /affect their activity. However, when the concept ‘short chain fatty acids’ is used people normally think of acetate, butyrate and propionate usually derived from bacteria, which specifically are sensed by the FFA2 and FFA3 receptors. So please avoid the use of SCFAs in relation to the HCA receptors throughout the paper. 2) More importantly it is today misleading towards readers – and editors - to associate HCA-2 with dyslipidemia (as also done by Yang et al). Yes, HCA2/GPR109A was originally orphanized as a receptor for the old drug ‘niacin’ back in 2003. This made the whole pharmaceutical industry search for novel HCA-2 agonists to become new and better drugs to treat dyslipidemia. However, already in 2012 it was very convincingly demonstrated that niacin’s lipid efficacy was independent of both HCA-2 and its effect on free fatty acids (antilipolytic effect in adipose) by use of HCA-2 KO mice and – importantly - data from two clinical trials using selective and efficacious HCA agonist (Lauring et al Sci Transl Med – 2012, 22:148ra115). Thus, HCA-2 is only mediating the flushing side effect of niacin and NOT its beneficial effects on lipids. 3) HCA-2 agonists do inhibit lipolysis, but they failed in clinical phase-2 as treatment of diabetes due to development of tolerance (GSK256073– Eur. J. Pharmacol 2015). 4) Good News, which instead can – and is - be used for seduction of readers and editors: HCA-2 is a very interesting target for treatment of inflammatory diseases, in particular neuroinflammation due to its expression on e.g. neuroprotective macrophages (e.g. Rahman et al. Nat.Comm 2014) and the HCA-2 agonist monomethylfumarate is a drug used for treatment of MS.

Some Minor points:

1. The agonists in the HCA receptors do not bind ‘shallow’. Their main ‘anchor point’ R111 (3.36) is in fact located one helical turn deeper than the corresponding, iconic anchor point for catecholamines in their receptors D(3,32).

2. line 230 – ICL1 should be ECL1

3. line 243 ‘suggesting’ should be ‘in agreement with the fact that’ or something like that.

4. Supplementary Fig 12b. The dark purple curve, which in fact is the only curve seriously shifted to the right is labeled R128A but should be labeled R218A. R128A is apparently an orange curve.

Reviewer #3 (Remarks to the Author):

This manuscript reported the active structures of HCA2 and HCA3 in complex with different ligands, as well as their molecular mechanism. The entry of lipid ligands into the binding pocket from the membrane, rather than the extracellular region, was also discussed. This finding is important for the research of lipid receptors with similar properties.

However, there are several areas that need improvement to make the conclusions sound. In addition, another work on HCA2 structures, including inactive and active conformations, was reported recently (Nat Commun 14, 1692 (2023)). Although it does not affect the independency of the work presented in this manuscript, we may expect some new advancement compared to the published work. My comments, advices and questions are listed below:

1. General comments about the manuscript: The introduction could benefit from additional information and greater clarity. Like in line 52, I'm not so sure whether GSK256073 was an agonist or antagonist when reading this part. The paper should also include more information about SUCR and the relationship between HACs and other lipid receptors in the δ branch. Additionally, the discussion session may need to be better organized. It was too much detail but difficult for readers to get the most important point, such as the key value to the related field and future perspectives that may be inspired by this study.

2. There are numerous mutation experiments in this manuscript, but the expression levels of the mutated constructs are lacking. It would be beneficial to include data on the key mutations, such as R1113.36, S179ECL2, Y284ECL2, R2516.55, L1584.56, W1895.38 and else, to demonstrate that the mutations do not significantly affect expression levels. Without this information, the conclusions may not be convincing. Such data can be well organized into a supplementary table.

3. In line 106-107, "Interestingly, R3.36 is conserved only in the HCA family, including OXER and GPR313 (Supplementary Fig. 9)." This sentence is quite confusing, does it mean OXER and GPR31 belong to the HCA family? Also, I didn't see GPR31 in Supplementary Fig.9. This should be added.

4. In Supplementary Fig.8e, the S179A appeared to improve the EC50 and Emax. While S179 was essential for the ligand binding, what's the possible explanation for this result?

5. Since there are no inactive structures for HCAs, SUCR and predicted AF2 model were used for discussing activation mechanisms of HCAs. The conclusions from comparing with AF2 model may not be convincing. Also I was not convinced by the "high sequence similarity" with SUCR. Sequence comparison

on the region critical for activation between HCAR and SUCR could be included in the figure to make the conclusion more reliable.

6. In the Gi coupling part, it is unclear which structure was used to represent active HCA. Was the Gi coupling interface of four structures exactly the same?

7. The discussion on the ligand entrance section was interesting and may provide one important highlight of this paper, however the experimental support to this conclusion is not solid. Since the W188A and H189A mutation experiment did not provide convincing evidence for the ligand entry from the membrane. Was there any other mutations or experiments which can fully block the space between TM4 and TM5?

8. The scale bars in Supplementary Fig.2b&e, 3b&e, 4b should be indicated for their length.

9. Line 129 in Supplementary Fig.15 legend, I think the “antagonist” should be “agonist”.

10. The whole manuscript needs more work to polish the text. Due to grammatical errors and unclear logical structure, some sentences are difficult for readers to understand. Just to give one example in line 45-46: HCA2 was first identified as a receptor for niacin and is the most well-studied pharmacologically. Also, from line 251-256, the N-terminus beta-hairpin and C18-C19 disulfide bond repeated twice.

Reviewer #4 (Remarks to the Author):

This manuscript describes structural analyses of agonist occupied hydroxycarboxylic acid receptor subtype 2 and 3 together with mutagenesis, binding/activation data and molecular dynamics stimulation. The work provide molecular insight into ligand binding and activation of HCA2 and HCA3 receptors and close family member. The authors selected to solve the cryoEM structures of human HCA2 heterotrimeric Gi complexes bound to the selective full agonist GSK256073, MK6892, partial agonist LUF6283 and non-selective full agonist acifran. To understand further agonist binding/activation mechanism, the acifran was also solved with complexes of human HCA3 heterotrimeric Gi. The structural analysis was complemented extensive mutation analysis of agonists binding and probe Gi protein activation using functional assays and molecular dynamics.

The work is carefully performed, well presented, and provides useful and noteworthy information for the field of GPCRs and short-chain fatty acid signalling receptors, and their ligand binding interactions, and more generally their potency as therapeutic targets towards dyslipidaemia, niacin flush and other autoimmune diseases. The work is very well suited for wide audience of Nature communications.

I have no major comments flaws in the data analysis, interpretation and conclusions regarding this manuscript. The methodology meets the standard usage in structural biology, the produced structures meet the validation criteria, data binding/functional data related mutagenesis and interpretation is well preformed, and sufficient detail are provided in the methods for the work to be reproduced.

Minor points

Active conformation of HCAs, lines 157-178

It is not clear to me, how to results suggest that HCAs have unique regulation mechanism among Gi/o coupled reported so far. What is described is very similar binding mode for Gi/o. Maybe rewrite and clarify the paragraph. The lower part of paragraph could be joined the next paragraph "Gi coupling interface"

The authors constructed inactive model of HCA2 using alphafold and compared it to active-state to further understand the helical changes during activation. Maybe using experimentally determined inactive-state structures of succinate or other close by family members could be used instead of the alphafold model?

Point-to-Point Response to the Reviewers

Manuscript ID; NCOMMS-23-08315

We sincerely appreciate the editor's and reviewers' valuable and insightful comments on our manuscript. We revised and improved our manuscript as suggested, and provide a point-to-point response to address each comment and question. Portions of the manuscript that were updated in response to the reviewers' comments are highlighted in the text.

REVIEWER COMMENTS

Reviewer #1 (Remarks to the Author):

The paper by Suzuki et al entitled "Structural basis of signaling mechanisms of 2 hydroxycarboxylic acid receptors through ligand binding" describes the ligand binding mechanism of the HCA2 and HCA3 receptors with multiple agonists. The authors performed cryo-EM and tested their structural findings using mutagenesis and MD simulations. This is the first study describing the structures of the HCA receptors and it provides a valuable framework for understanding ligand binding and activation. While this is a really good study with gorgeous cryo-EM maps at pretty high resolution that are supported by extensive mutagenesis experiments, I have significant concerns about the quality of the models. While in itself it does not affect the conclusions of the paper, they are quite serious and need to be addressed before the publication and before the coordinates are available to the general public.

Response:

We thank the reviewer for these positive and very clear specific comments as well as the valuable advice about PDB models. For PDB IDs 8IHH, 8IHI, 8IHJ, and 8IHK, we found that the pixel size was incorrect. Consequently, calibration was carried out to rectify this issue. We have also refined the models and updated the statistics table (Supplementary Table 1) accordingly.

Supplementary Table 1

Cryo-EM data collection, refinement, and validation statistics

	HCA2- GSK256073 (EMDB- 3542) (PDB 8IHB)	HCA2- MK6892 (EMDB- 35443) (PDB 8IHF)	HCA2- LUF6283 (EMDB- 35444) (PDB 8IHH)	HCA2- acifran (EMDB- 35445) (PDB 8IHI)	HCA3- acifran (EMDB- 35446) (PDB 8IHJ)	HCA3- acifran (local) (EMDB -35447) (PDB 8IHK)
Data collection and processing						
Magnification	105K	105K	50K	50K	60K	

Voltage (kV)	300	300	300	300	300	
Electron exposure (e-/Å ²)	49	49	69.6	69.6	71.2	
Defocus range (µm)	-0.7 to -1.5	-0.7 to -1.5	-1.0 to -2.0	-1.0 to -2.0	-1.0 to -2.0	
Pixel size (Å)	0.83	0.83	0.99	0.99	0.78	
Final particle images (no.)	245,517	181,958	181,273	146,577	95,567	
Map resolution (Å)	2.85	2.97	3.13	3.17	3.18	3.33
FSC threshold	0.143	0.143	0.143	0.143	0.143	0.143
Map resolution range (Å)	2.8-4.8	2.8-4.8	2.8-4.8	2.8-4.8	2.8-4.8	2.8-4.8
Refinement						
Model resolution (Å)	3.0	3.1	3.2	3.2	3.3	3.6
FSC threshold	0.5	0.5	0.5	0.5	0.5	0.5
Model resolution range (Å)	n/a	n/a	n/a	n/a	n/a	n/a
Map sharpening B factor (Å ²)	-96.8	-97.4	-87.8	-87.3	-94.5	-100.4
Model composition						
Non-hydrogen atoms	8709	8702	8649	8672	8510	2253
Protein residues	1131	1131	1131	1131	1122	283
Ligands	1	1	1	1	1	1
R.m.s. deviations						
Bond lengths (Å)	0.004	0.003	0.002	0.003	0.002	0.002
Bond angles (°)	0.6151	0.495	0.519	0.522	0.463	0.582
Validation						
MolProbity score	1.59	1.59	1.60	1.44	1.50	1.36
Clashscore	6.50	5.98	6.35	5.72	5.55	3.56
Poor rotamers (%)	0.11	0.11	0.11	0.22	0.11	0.42
Ramachandran plot						
Favored (%)	96.50	96.14	96.32	97.31	96.75	96.44
Allowed (%)	3.50	3.86	3.68	2.69	3.25	3.56
Disallowed (%)	0	0	0	0	0	0

The major issues with the PDB models are outlined below.

1. Rotamers:

a. For all models there is an extremely low number of favored rotamers (e.g. ~70% for 8IHK, 65% for 8IHB (and others too) when this should be closed to 98%).

Response:

Thank you for your comments. We carefully checked and refined our models. Now, the number of favored rotamers is close to 98% for all models.

b. Number of rotamer outliers is also too high for 8IHB and 8IHF considering the high quality of the maps

Response:

We carefully checked and refined the models. In the current models, the rotamer outliers have the values shown in the following Table (The value of the output in Phenix).

8IHB	8IHF	8IHH	8IHI	8IHJ	8IHK
0.11	0.11	0.11	0.22	0.11	0.42

2. Clashscores

a. Very high clash scores for 8IHH, 8IHI, 8IHJ and 8IHK.

Response:

In the current models, the clashscores have the values shown in the following Table (The value of the output in Phenix).

8IHB	8IHF	8IHH	8IHI	8IHJ	8IHK
6.50	5.98	6.35	5.72	3.55	3.56

b. Numbers in Table 1 do not correlate to the Molprobit numbers – e.g. 12.45 for clashscore for 8IHK vs 8.92 in the table 1

Response:

Thank you for pointing this out. We checked it carefully. The refined model for 8IHK shows that the Molprobit number is 1.36 and the clash score is 3.56.

3. Ramachandran

a. Very suspicious distribution of -3.42 ± 0.43 for 8IHK (possibly others as I havent checked all the models) suggesting that the Ramachandran angles were forced into the “favoured” space without consulting the residue environment

Response:

Thank you for your suggestions. We carefully checked and refined our models. Please see below for a summary of the statistics regarding the Ramachandran plot Z-score (The value of the output in Phenix).

8IHB	8IHF	8IHH	8IHI	8IHJ	8IHK
-0.69±0.24	-0.40±0.25	-0.7±0.25	-0.60±0.25	-0.9±0.25	-0.90±0.49

b. Favoured Ramachandran – too low for 8IHF and 8IHK

Response:

We carefully checked and refined the models. The refined models’ statistics are shown as follows.

Ramachandran plot	8IHB	8IHF	8IHH	8IHI	8IHJ	8IHK
-------------------	------	------	------	------	------	------

Favoured (%)	96.50	96.14	96.32	97.31	96.75	96.44
Allowed (%)	3.50	3.86	3.68	2.69	3.25	3.56
Disallowed (%)	0	0	0	0	0	0

c. Please switch out which are favoured, allowed and disallowed in Table 1 (this is minor – just a typo)

Response:

We apologize for the typo. It has been corrected.

4. Ligands

a. Wrong stereochemistry of GSK256073 – around the N atom – needs to be planar – then the lipid tail will actually fit

b. LUF6283 – doesn't fit to the density well

Response:

Thank you for your comment; this has been corrected (Figure 1 a, b for revision).

a

b

Figure 1 for revision; Ligand and density map

Superimposed ligand and density map. Ligands are shown in stick representation. cryo-EM density is represented as a transparent surface. GSK256073 (a), LUF6283 (b)

5. Why are coordinates different between the receptor in 8IHJ and 8IHK?? They are the same receptor and the same map (only local vs. global.)

Response:

We carefully checked the 2 maps and the coordinates are identical. Please confirm this.

Minor modelling issues:

6. MMK6892- check NAG stereochemistry/conformation (maybe check other models for the same issues as I haven't check all of them)

Response:

We carefully checked all models and corrected them, thank you.

7. All structures – add a carboxylic acid to the end of Galpha

Response:

We thank the reviewer for these comments. As suggested, we added a carboxylic acid to the end of G α .

Comments about the manuscript itself:

1. While this is a structural biology paper and structures are the main point of this study, to me it felt that there is an overwhelming amount of structural discussion that is targeted to a very specialist audience (discussion about receptor activation, G protein coupling interfaces etc.). This amount of information and detail makes this for a very difficult read, and at times the point authors are making (e.g. about the selectivity and others) gets lost. Perhaps some of this discussion can be shortened and/or moved to a supplement?

Response:

As the reviewer suggested, the discussion of G α selectivity is very specialized and may not be suitable for general readers. We decided to focus our manuscript mainly on ligand selectivity and activation of the HCA receptor. Therefore, the following text on G protein selectivity was written only briefly.

Line 96-97 Unclear why authors compare HCA receptor binding site to that of the amine-coupled receptors.

Response:

HCA2 is activated by small ligands such as niacin. We showed how the ligand-binding pocket of HCA2 differs from that of a typical GPCR, such as monoamine receptors, which are the most well-studied in terms of ligand recognition. In fact, the ligand-binding pocket of the monoamine receptor is filled by aromatic residues in HCA2. To address the reviewer's point, we clarified this in the manuscript as follows.

page 6,line 108

All agonists, except for MK6892, had a very similar binding pose in the orthosteric site. As the ligand binding pocket of HCA2 is formed by TM1, TM2, TM3, TM7, and extracellular loop 2 (ECL2) (Supplementary Fig. 7a), the binding site differs significantly from those of well-studied class A monoamine-coupled receptors such as serotonin²⁰ and dopamine receptors²¹.

3. Line 158 “the inactivated AlphaFold2 (AF2) structure” perhaps authors meant “inactive”?

Response:

Yes, we apologize for this mistake and changed this part. We avoided using the inactive AF2 structure for comparison and instead used the already reported the δ -branch GPCRs for comparison, as mentioned later in response to Reviewer #4.

4. Line 240 “The aromatic ring of acifran forms a hydrophobic interaction with W91 in HCA2” – unclear how is this happening as W91 is $>4\text{\AA}$ away?

Response:

We thank the reviewer for these comments. We apologize for the inaccurate and incorrect descriptions that the aromatic ring of acifran forms hydrophobic interactions with W91 in HCA2. As the reviewer pointed out, W91 is more than 4 \AA away from the aromatic ring of acifran, thus it was a van der Waals interaction, not a hydrophobic interaction. According to the reviewer's comments, we revised the text as follows.

page 7, line 127

The acyl tails of GSK256073 and LUF6283, and the aromatic ring of acifran are surrounded by several hydrophobic residues (L83^{2.60}, W91^{ECL1}, M103^{3.28}, L107^{3.29}, C177^{ECL2}, F180^{ECL2}, and F267^{7.35}), which form hydrophobic and van der Waals interactions with them, and fit into the ligand-binding pocket.

5. Line 253 “suggesting that the affinity of acifran is approximately 10 times higher for HCA2 than for HCA3” -> perhaps authors meant “resulting in”?

Response:

We thank the reviewer for this comment. Reviewer #2 pointed out the same thing, and we changed "suggesting..." to "consistent with the 10-fold decrease in affinity" as follows.

Page 12, line 254

On the other hand, in HCA3, pocket I cannot stably hold the aromatic ring of acifran due to the larger volume (Fig. 5f), consistent with the 10-fold decrease in affinity².

Reviewer #2 (Remarks to the Author):

In the present paper Suzuki and coworkers present four novel cryo-EM structures of the hydroxycarboxylic acid receptor 2 (HCA-2), in complex with heterotrimeric Gi and four different agonist molecules: 1) the 40-year old niacin analog Acifran (used for dyslipidemia), 2) the partial agonist LUF6283, 3) the high affinity, and highly selective GSK256073 (having been in phase-2 clinical trials for diabetes); and 4) the high affinity, highly selective, late stage Merck agonist MK6892. The authors also present the first cryo-EM structure of the closely related HCA-3 – only found in humanoids - in complex with heterotrimeric Gi and the dual specific Acifran agonist. The structural analysis is complemented by a comprehensive mutational and signal transduction analysis of a long list of residues involved in ligand or G protein binding and receptor activation in both receptors. The active HCA2 cry-EM structures are compared to an X-ray structure of the inactive form of a related receptor, SUCNR1 and to an Alpha-fold generated model of inactive HCA2 itself.

This paper constitutes an amazing amount of solid structural biology work and information plus carefully performed molecular pharmacological work on two interesting/important receptors, of which HCA-2 – originally known as the niacin receptor - has been a major drug target in the pharmaceutical industry aiming at changing medical indications over the years: dyslipidemia, diabetes, neurological inflammation etc. and consequently a rich pharmacology to characterize as here done very elegantly.

Unfortunately, a recent publication also in Nature Comm. by Yang and coworkers appears to steal some of the thunder of this story. However, that is only from a superficial point of view as they only have HCA-2 in complex with MK6892 (basically identical to the present structure) plus a mutationally ‘frozen’ inactive HCA-2 structure and consequently are only able to tell a rather simple story of yet another ligand bound in a GPCR. Yang and coworkers are, due to their lack of the three other ligand-receptor complexes, unable to tell and address the interesting and novel story of how polar ligands in HCA receptors get access to a deep pocket, which is totally secluded from the extracellular aqueous phase. That is, a story which is addressed in the present version of this paper, but which result-wise has been hidden in the supplementary material and only is told in relatively few lines in the Results and Discussion.

Response:

We very much appreciate the positive comments and scientifically important feedback.

We responded to the suggestions point-to-point and revised the manuscript as follows.

Major Point:

1. The authors should simply restructure the paper to make their proposed novel ligand entry the major theme. The data presented in supplementary Fig 15 should be included in the main paper and presented and discussed in much more detail. You should focus on the story of the positively charged lateral entry path for these anionic, polar ligands from the lipid head group area as the main novel message of the paper. That is, instead of just another long presentation and discussion of ligand binding in an orthosteric binding pocket and a similar story of the receptor G protein interactions. Neither of these stories providing significant novel information – just as they did not in the Yang paper and just as they do not in most of the many cryo-EM GPCR papers which currently are being published. Your paper can be truly different by e.g. telling that: 1) It is structurally impossible for both the polar endogenous ligands (see point #2) as well as for the synthetic agonists to enter the orthosteric binding site ‘from the top’ as non-lipid ligand normally do in GPCRs; 2) All four HCA-2 agonist are firmly bound deeply below a totally ‘locked lid’ interacting with the crucial R3.36 (see minor point concerning deepness) ; 2) The ‘lid’ is composed of ECL-2 and the N-terminus of the receptor ‘knitted together’ by three disulfide bonds which already are mutationally addressed in your paper. 3) You have identified

a lateral entrance channel between TM-IV and-V and demonstrated how it mouth and surrounding are electropositively charged 4) importantly, although e.g. GSK256073 occupies the central deep part of the orthosteric pocket it is nevertheless affected by mutations around the proposed lateral entry site, i.e. ‘far’ away from its final binding site. 5) Finally, the discussion (line 322-327) of the fact that receptor selectivity towards different types of ligands e.g. aromatic D amino acids surprisingly cannot be explained by differences in the orthosteric binding pocket, could very likely instead be explained by differences in the entry path as demonstrated convincingly by MD simulations in the catecholamine receptors. You should therefore at least compare the structure of the proposed entry path of HCA2 with that of HCA-3 – they are likely different.

Response:

We thank the reviewer for the constructive advice. We are happy to reorganize our manuscript according to the reviewer’s comments. As for comments 1)~4), we would like to move Supplemental Fig. 15 to Fig. 4 in the main manuscript and describe the ligand entries in detail as suggested. Unfortunately, however, the evidence seems insufficient to make the ligand pathways a major theme. Therefore, we kept ligand recognition and selectivity as the main themes in our paper.

As for comment 5), we apologize for our incorrect description of the selectivity for “D-amino acid” in the Discussion, “HCA3 is activated by several D-amino acids⁶³, but these amino acids do not activate HCA2. As the difference in pocket size is insufficient to explain these findings, other factors may be associated with the ligand selectivity of the 2 receptors”. HCA3 is indeed activated by D-(+)-phenyl lactic acid and DL-indole-3-lactic acid, but not by D-amino acids. Therefore, the selectivity of HCA3 to those lactic compounds might rather be explained by their large molecular weights and high hydrophobicity, while the residues at the ligand entry points are conserved in HCA2 and HCA3, making it likely that all the ligands pass through the same pathways. We corrected the description of the citation (Peters et al (2019)) in the revised manuscript.

In response to the comments, we revised many parts of the text on page 7, Fig. 4, and page 10.

page 10, line 194

HCA2 ligand entrance

The structure of HCA2 shows that ECL2 on the extracellular side has a conserved β -hairpin structure that passes over the ligand-binding pocket and connects to TM5 (Fig. 4a). C177^{45,50} of ECL2 forms a disulfide bond with C100^{3,25} of TM3, which is conserved in many class A GPCRs and is essential for the formation of the ligand binding pocket^{46,47}. The N-terminus of HCA2 also forms a β -hairpin structure, in which C18 and C19 form disulfide bonds with C266^{7,25} and C183^{5,33}, respectively (Fig. 4a). Mutations of C100^{3,25} and C177^{45,50} cause a loss of activity and significantly reduce receptor surface expression levels, suggesting that the 2 cysteines are essential for the receptor folding and trafficking²⁰. Mutations of C266^{7,25} and C183^{5,33} significantly reduce receptor activity but do not affect membrane localization²⁰, however, suggesting a crucial role of the additional disulfide bonds in proper ligand pocket construction.

On the basis of our structure, the agonist of HCA2 is completely occluded in the ligand pocket (Fig. 4a). It is structurally unfavorable for both the polar endogenous ligands as well as for the synthetic agonists to enter the orthosteric binding site ‘from the top’ as non-lipid ligands normally do in GPCRs. This means that in contrast to receptors that bind ligands from the extracellular side, such as amine-coupled receptors, ligand entry to the

binding pockets of HCA2 would be from the lipid membrane through the lateral gate. Investigation of the charge properties in the extracellular region of HCA2 revealed the presence of a highly basic region located at TM4 and TM5 (Fig. 4b). Nicotinic acid, β -hydroxybutyrate, and β -hydroxyoctanoic acid are negatively charged in the endogenous environment (Fig. 4c). Considering the extended ligand-binding pocket in the MK6892-bound structure, we hypothesize that the gap between TM4 and TM5 is the entrance to the ligand pocket. Therefore, we focused on H188 and W189, which are located at the possible ligand entrance (Fig. 4d). We speculate that alanine mutants of those residues are unable to completely close the ligand entrance and thus reduce the ligand response. In support of our hypothesis, W188A and H189A decreased the potency, even though these residues are distant from the binding site of GSK256073, LUF6283, and acifran (Fig. 4e). To assess the functional role of the potential ligand entrance, we performed MD simulations of HCA2, with and without GSK256073. The distance between L158^{4.56} and W189^{5.38} was essentially the same as that of the initial structure for about 900 ns in the presence of GSK256073, whereas under ligand-free conditions, the distance increased (Supplementary Fig. 13a, b). This observation may indicate that the upper half of TM5 itself has higher flexibility without ligand binding. The finding that the alanine mutants for H189 and W188 did not affect the activity of MK6892 is possibly due to the broad interaction of MK6892 with HCA2 (Fig. 4e). These results suggest that the space between TM4 and TM5 may be the ligand entrance in HCA2, and endogenous agonists and synthetic agonists insert into the ligand pocket from the lateral gate within the lipid bilayer membrane.

Fig. 4 | Ligand entrance of HCA2

a Top view of the agonist-bound HCA2 in cartoon representation and transparent surface representation. The extracellular loops are colored individually (N-terminus, cyan; ECL1, green; ECL2, orange; ECL3, pink). Six cysteine residues forming disulfide bonds are shown in stick representation. **b** Electrostatic potential surface of HCA2 at the extracellular side, ranging from -10 kT/e (red) to $+10$ kT/e (blue). Residues around the ligand entrance are labeled and shown in stick representation. **c** Two-dimensional representation of chemical structures of nicotinic acid, β -hydroxybutyrate, and β -hydroxyoctanoic acid. **d** Structural comparison in the cross-sectional views of the cryo-EM densities of HCA2 bound with GSK256073 (upper left), LUF6283 (upper right), acifran (under left), and MK6892 (under right). Models of the agonist are shown in stick representation with their densities omitted to show the ligand binding pockets more clearly. **e** Concentration-response curves of cAMP inhibition assay for W188A and H189A mutants.

page 15, line 327

HCA2 and HCA3 have different ligand selectivities despite having more than 90% sequence identity. The difference in the amino acid sequence between the 2 receptors is 15 residues except for the C-terminus. We showed that the 6 amino acids (positions 83, 86, 91, 103, 107, 178) contribute primarily to the volume and shape of the ligand-binding pockets. The volume of sub-pocket I of HCA2 is smaller than that of HCA3, favoring smaller molecules such as niacin and β -hydroxybutyrate, whereas 3-hydroxyoctanoic acid, an endogenous agonist of HCA3, has a longer carbon chain than β -hydroxybutyrate, an endogenous HCA2 agonist. The N86Y/M103V/L107F triple mutant of HCA2 is reported to increase its activity by β -hydroxyoctanoate to levels comparable to those of HCA3³, which can be explained by the extended sub-pocket I. HCA3 is also activated by lactic acid bacteria-derived metabolites such as D-PLA and ILA⁴⁹, but these amino acids do not activate HCA2. These compounds are slightly larger and more hydrophobic, which may favor HCA3. The MK6892-bound structure, however, shows that the ligand-binding sites in GPCRs are highly flexible. More structures of HCAs in complex with various ligands will provide clues to elucidate the ligand selectivity and potentially drive more efficient structure-based drug design.

2. An in-depth MD analysis of ligand entry and movements of residues in and around the entry channel would be really great to see. But it would be a huge study which will be more suited as a follow up study. The current MD simulations or rather the displayed data (Fig. 6D) concerning the movements of L258 relative to W188 with and without the GSL ligand in place are not very convincing. It is possible that other measures such as rotation of W188, which potentially could function as a 'swing door' in the entry channel would give more interesting and significant differences?

Response:

Thank you for your insightful suggestions. According to this suggestion, we tested other measures such as the rotation of W188 (Figure 2 for revision). The result was almost consistent with the previous Fig.6d (now Supplementary Fig 13b) in which the distance between L158 of TM4 and W188 of TM5 was measured. Therefore, we concluded that the result in Fig. 6d (now, Supplementary Fig. 13b) is plausible as a measurement method. Consistent with this result, we confirmed a significant change in TM5 compared with the reported structure of the inactivated HCA2 (Supplementary Fig. 15a, b). These observations indicated that the opening and closing of the ligand entrance is caused by

conformational changes in the upper half of TM5 rather than by only specific residues such as W188 and H189.

Nevertheless, we decided to move the results to a supplementary figure, as the results are too weak to argue and are thus not suitable to discuss in the main figure.

Figure 2 for revision; Rotation measures of W188 during 900-ns MD simulation
Molecular dynamics (MD) simulations in the presence (left) and absence (right) of GSK256073. Temporal changes in the angle of W188 (left) and absence (right) of GSK256073 are shown. Simulations were performed in 3 independent runs of the presence and absence of GSK266073, respectively.

3. The whole background narrative needs to be seriously changed. 1) HCA receptors are not ‘short chain fatty acid receptors’. Their endogenous agonists are as their name tells, hydroxycarboxylic acids (lactate, β -hydroxybutyrate/ketone bodies, and β -hydroxyoctanoate). Some classical short chain fatty acids may also cross react with them /affect their activity. However, when the concept ‘short chain fatty acids’ is used people normally think of acetate, butyrate and propionate usually derived from bacteria, which specifically are sensed by the FFA2 and FFA3 receptors. So please avoid the use of SCFAs in relation to the HCA receptors throughout the paper. 2) More importantly it is today misleading towards readers – and editors – to associate HCA-2 with dyslipidemia (as also done by Yang et al). Yes, HCA2/GPR109A was originally orphanized as a receptor for the old drug ‘niacin’ back in 2003. This made the whole pharmaceutical industry search for novel HCA-2 agonists to become new and better drugs to treat dyslipidemia. However, already in 2012 it was very convincingly demonstrated that niacin’s lipid efficacy was independent of both HCA-2 and its effect on free fatty acids (antilipolytic effect in adipose) by use of HCA-2 KO mice and – importantly - data from two clinical trials using selective and efficacious HCA agonist (Lauring et al Sci Transl Med – 2012, 22:148ra115). Thus, HCA-2 is only mediating the flushing side effect of niacin and NOT its beneficial effects on lipids. 3) HCA-2 agonists do inhibit lipolysis, but they failed in clinical phase-2 as treatment of diabetes due to development of tolerance (GSK256073– Eur. J. Pharmacol 2015). 4) Good News, which instead can – and is - be used for seduction of readers and editors: HCA-2 is a very interesting target for treatment of inflammatory diseases, in particular neuroinflammation due to its expression on e.g. neuroprotective macrophages (e.g. Rahman et al. Nat.Comm 2014) and the HCA-2 agonist monomethylfumarate is a drug used for treatment of MS.

Response:

Thank you for your informative comments; we apologize for the misleading sentences. According to suggestions (1)-(3), we removed the phrases concerning SCFAs and dyslipidemia from the manuscript. 4) Thank you for letting us know the important information about the interesting role of HCA2 as a drug target for the treatment of

inflammatory diseases. According to your suggestion, we modified the Introduction as follows.

page 3, line 33

Introduction The hydroxycarboxylic acid receptor (HCA) family consists of the typical metabolism-sensing receptors present in humans and belongs to the class A GPCRs. The HCA family comprises 3 subtypes, HCA1 responding to lactate¹, HCA2 responding to niacin and hydroxybutyrate (BHB)², and HCA3 responding to 3-hydroxyoctanoic acid³, and signals through the inhibitory Gi/o family of G proteins⁴. Downstream signaling is diverse and tissue-dependent. Among these subtypes, HCA2 is predominantly expressed in the intestines and white/brown adipocytes, as well as in various immune cells, including dendritic cells, monocytes, macrophages, neutrophils, and epidermal Langerhans cells⁵⁻⁸. Therefore, HCA2 is involved in many pathophysiological processes. Recent studies demonstrated that potent drugs acting on HCA2 could have beneficial effects on multiple neurologic diseases. For example, an FDA-approved formulation of niacin, Niaspan, stimulates a broad and complex protective response mediated by microglia, leading to a lower plaque burden, reduced neuronal loss, and improvement in working memory deficits⁹. BHB, a ketone body, also induces a neuroprotective phenotype in bone marrow-derived macrophages invading the brain and acts as an endogenous factor that protects against stroke and neurodegenerative diseases, and this action is mediated by HCA2⁵. Several potent HCA2 agonists, including niacin-containing acipimox, acifran, and monomethyl fumarate (MMF), are currently approved for the clinical treatment of cardiovascular and neurologic diseases. Niacin is being investigated as a treatment for Parkinson's disease¹⁰ and glioblastoma due to its immunomodulatory and neuroprotective properties and is currently undergoing clinical trials (NCT03808961, NCT04677049). The niacin-derived antiphlogistic agents acipimox and acifran are commonly used clinically to treat dyslipidemia and atherosclerosis^{11,12}. In addition, MMF was approved by the FDA in 2020 for the treatment of relapsing-remitting multiple sclerosis¹³. Experimental evidence shows that MMF activates HCA2, resulting in a change in microglia from a pro-inflammatory form to a neuroprotective form. Thus, HCA2 is an attractive drug target for a very diverse range of diseases, but there are several problems with the molecule. These drugs cause severe flushing (known as the niacin flush), which decreases patient compliance. Therefore, much effort has been focused on developing alternatives associated with less flushing. LUF6283, belonging to the pyrazole class of compounds, is a partial agonist with lower affinity¹⁴. LUF6283 achieves the action of niacin without the undesirable flushing side effects. In addition, the high-affinity HCA2-selective agonists MK-6892¹⁵, SCH900271¹⁶, and GSK256073¹⁷ were developed. GSK256073 binds both HCA2 and HCA3 but is 100-fold more selective for HCA2. MK6892 is an HCA2-selective agonist with a chemical structure that differs from the 3 preceding compounds. The lack of structural information for the active state of any of the HCA subtypes as well as the lack of a structural framework for HCA-ligand binding and selectivity, however, substantially impede advances in rationale drug discovery.

HCA3 (GPR109b) exists only in hominids, including humans. Despite the high sequence similarity between HCA2 and HCA3, their endogenous ligands differ. Pharmacologic and computational analyses indicate that amino acids in the extracellular half of the transmembrane (TM) domain are responsible for the different ligand preferences^{18,19}. How differences in the ligand binding pockets determine ligand preference, however, is still unclear. Structural information on HCA2 and HCA3 will provide important clues for the development of subtype-selective drugs.

To investigate the molecular mechanisms underlying HCA ligand recognition and activation, we determined the cryo-EM structures of human HCA2-Gi complexes bound to

the HCA2-selective full agonists GSK256073 and MK6892, the partial agonist LUF6283, and the nonselective synthetic full agonist acifran. For a deeper understanding of HCAs, we also analyzed the structure of the human HCA3-Gi complex with acifran. These structural and mutational analyses and molecular dynamics (MD) simulation experiments reveal the molecular basis for understanding how HCA recognizes ligands and activates G proteins.

Some Minor points:

1. The agonists in the HCA receptors do not bind ‘shallow’. Their main ‘anchor point’ R111 (3.36) is in fact located one helical turn deeper than the corresponding, iconic anchor point for catecholamines in their receptors D(3,32).

Response:

We thank the reviewer for these important comments. We apologize for the inaccurate descriptions regarding agonist binding to HCA receptors. We removed the term “shallow” in our manuscript.

2. line 230 – ICL1 should be ECL1

Response:

Thank you for pointing out this typo, which we have corrected in the revised manuscript (line 128).

3. line 243 ‘suggesting’ should be ‘in agreement with the fact that’ or something like that.

Response:

Thank you for your suggestion. We changed the phrase “suggesting...” to “consistent with the 10-fold decrease in affinity” as follows.

Page 8, line 254

On the other hand, in HCA3, pocket I cannot stably hold the aromatic ring of acifran due to the larger volume (Fig. 5f), consistent with the 10-fold decrease in affinity².

4. Supplementary Fig 12b. The dark purple curve, which in fact is the only curve seriously shifted to the right is labeled R128A but should be labeled R218A. R128A is apparently an orange curve.

Response:

Thank you for pointing this out. We changed the label “R128A” to “R218A”.
Supplementary Fig. 12b has been changed to Fig. 6f.

Fig. 6 | The G protein interfaces of HCA2

a Interface between the active form of HCA2 (cartoon representation) and the bound Gi protein (surface representation).

b Hydrophobic interactions between HCA2 and the C terminus of the Gi subunit. **c**

Interactions between ICL2 and a hydrophobic cleft formed by Gi. **d** Interactions of ICL3 with Gi. **e** Interactions of ICL1 with the C-terminus of Gi. Hydrophilic interactions are indicated by blue dashed lines. **f** Close-up view of the interface between HCA2 and Gi.

Side chains of positively and negatively charged amino acids are shown in stick representation. The electrostatic potential surface of Gi ranges from -10 *kT/e* (red) to +10 *kT/e* (blue). **g** cAMP inhibition assay of Gi protein binding interface mutants.

Reviewer #3 (Remarks to the Author):

This manuscript reported the active structures of HCA2 and HCA3 in complex with different ligands, as well as their molecular mechanism. The entry of lipid ligands into the binding pocket from the membrane, rather than the extracellular region, was also discussed. This finding is important for the research of lipid receptors with similar properties.

However, there are several areas that need improvement to make the conclusions sound. In addition, another work on HCA2 structures, including inactive and active conformations, was reported recently (Nat Commun 14, 1692 (2023)). Although it does not affect the independency of the work presented in this manuscript, we may expect some new advancement compared to the published work. My comments, advices and questions are listed below:

Response:

We thank the reviewer for these constructive comments and valuable feedback. In response to the comments, we revised the manuscript as follows.

1. General comments about the manuscript: The introduction could benefit from additional information and greater clarity. Like in line 52, I'm not so sure whether GSK256073 was an agonist or antagonist when reading this part. The paper should also include more information about SUCR and the relationship between HACs and other lipid receptors in the δ branch. Additionally, the discussion session may need to be better organized. It was too much detail but difficult for readers to get the most important point, such as the key value to the related field and future perspectives that may be inspired by this study.

Response:

Thank you for your constructive advice. We carefully revised the Introduction and organized the Discussion according to the comments.

page 3, line 33

Introduction

The hydroxycarboxylic acid receptor (HCA) family consists of the typical metabolism-sensing receptors present in humans and belongs to the class A GPCRs. The HCA family comprises 3 subtypes, HCA1 responding to lactate¹, HCA2 responding to niacin and hydroxybutyrate (BHB)², and HCA3 responding to 3-hydroxyoctanoic acid³, and signals through the inhibitory Gi/o family of G proteins⁴. Downstream signaling is diverse and tissue-dependent. Among these subtypes, HCA2 is predominantly expressed in the intestines and white/brown adipocytes, as well as in various immune cells, including dendritic cells, monocytes, macrophages, neutrophils, and epidermal Langerhans cells⁵⁻⁸. Therefore, HCA2 is involved in many pathophysiologic processes. Recent studies demonstrated that potent drugs acting on HCA2 could have beneficial effects on multiple neurologic diseases. For example, an FDA-approved formulation of niacin, Niaspan, stimulates a broad and complex protective response mediated by microglia, leading to a lower plaque burden, reduced neuronal loss, and improvements in working memory deficits⁹. BHB, a ketone body, also induces a neuroprotective phenotype in bone marrow-derived macrophages invading the brain and acts as an endogenous factor that protects against stroke and neurodegenerative diseases, and this action is mediated by HCA2⁵. Several potent HCA2 agonists, including niacin-containing acipimox, acifran, and monomethyl fumarate (MMF), are currently approved for the clinical treatment of cardiovascular and neurologic diseases. Niacin is being investigated as a treatment for Parkinson's disease¹⁰ and glioblastoma due to its immunomodulatory and neuroprotective properties and is currently undergoing clinical trials (NCT03808961, NCT04677049)¹¹.

The niacin-derived antiphlogistic agents acipimox and acifran are commonly used clinically to treat dyslipidemia and atherosclerosis^{12,13}. In addition, MMF was approved by the FDA in 2020 for the treatment of relapsing-remitting multiple sclerosis¹⁴. Experimental evidence shows that MMF activates HCA2, resulting in a change in microglia from a pro-inflammatory form to a neuroprotective form. Thus, HCA2 is an attractive drug target for a very diverse range of diseases, but there are several problems with the molecule. These drugs cause severe flushing (known as the niacin flush), which decreases patient compliance. Therefore, much effort has been focused on developing alternatives associated with less flushing. LUF6283, belonging to the pyrazole class of compounds, is a partial agonist with lower affinity¹⁵. LUF6283 achieves the action of niacin without the undesirable flushing side effects. In addition, the high-affinity HCA2-selective agonists MK-6892¹⁶, SCH900271¹⁷, and GSK256073¹⁸ were developed. GSK256073 binds both HCA2 and HCA3 but is 100-fold more selective for HCA2. MK6892 is an HCA2-selective agonist with a chemical structure that differs from the 3 preceding compounds. The lack of structural information for the active state of any of the HCA subtypes as well as the lack of a structural framework for HCA-ligand binding and selectivity, however, substantially impede advances in rationale drug discovery.

HCA3 (GPR109b) exists only in hominids, including humans. Despite the high sequence similarity between HCA2 and HCA3, their endogenous ligands differ. Pharmacologic and computational analyses indicate that amino acids in the extracellular half of the transmembrane (TM) domain are responsible for the different ligand preferences^{19,20}. How differences in the ligand binding pockets determine ligand preference, however, is still unclear. Structural information on HCA2 and HCA3 will provide important clues for the development of subtype-selective drugs.

To investigate the molecular mechanisms underlying HCA ligand recognition and activation, we determined the cryo-EM structures of human HCA2-Gi complexes bound to the HCA2-selective full agonists GSK256073 and MK6892, the partial agonist LUF6283, and the nonselective synthetic full agonist acifran. For a deeper understanding of HCAs, we also analyzed the structure of the human HCA3-Gi complex with acifran. These structural and mutational analyses and molecular dynamics (MD) simulation experiments reveal the molecular basis for understanding how HCA recognizes ligands and activates G proteins.

page 6, line 179

HCA2 and SUCR are classified in the δ branch of class A GPCRs, including P2Y1³⁹, P2Y12⁴⁰,

CysLT1/2^{41,42}, PAR1/2^{43,44}, PAFR⁴⁵, GPR35³¹, and LPA6⁴⁶. We investigated whether the conformation changes observed in comparison with SUCR are conserved features among class A GPCR family members by comparing the structures of the reported δ -branch GPCRs. Most class A GPCRs have Trp residues at position 6.48 in TM6, which recognize their ligands and initiate the conformational changes required for receptor activation, while the δ -branch receptors have Phe or Tyr residues instead of Trp at this position (Supplementary Fig. 12a). Focused on the conformational change near the microswitch upon agonist binding, the upper half of TM5 of HCA2 bends inward starting at conserved Pro^{5.50}, and upward shifts of TM3 are commonly observed in all the known structures of the δ -branch GPCRs (Supplementary Fig. 12b-i). The rotamer of F^{6.44} and F^{6.48} adopts a “downward” conformation in all the structures, and TM6, including these residues, in the active HCA2 is pushed outward. The conformation observed in HCA2 is also consistent with that of G13-bound GPR35 (Supplementary Fig. 12j), suggesting that other δ -branch receptors have similar activation mechanisms.

Supplementary Fig. 12 | Structural comparison of δ -branch GPCRs focused on the micro-switch

a Sequence alignment of amino acid residues focused on the conserved microswitch of δ -branch class A GPCRs. **B-j** Structural comparison of HCA2 (orange) with other δ -branch class A GPCRs (gray), including PAR1 (PDB: 3VW7), PAR2 (PDB: 5NDD), CysLT1 (PDB: 6RX5), CysLT2 (PDB: 6RZ6), P2Y1 (PDB: 4XNW), P2Y12 (PDB: 4PXZ), PAFR (PDB: 4PXZ), LPA6 (PDB: 5XSZ), and GPR35 (PDB: 8H8J)

2. There are numerous mutation experiments in this manuscript, but the expression levels of the mutated constructs are lacking. It would be beneficial to include data on the key mutations, such as R1113.36, S179ECL2, Y284ECL2, R2516.55, L1584.56, W1895.38 and else, to demonstrate

that the mutations do not significantly affect expression levels. Without this information, the conclusions may not be convincing. Such data can be well organized into a supplementary table.

Response:

We appreciate the suggestion for the additional experiments. We completely agree that we should check that the mutations do not significantly affect the expression levels. We measured the cell surface expression levels of the HCA and HCA3 mutants using ELISA and summarized the data as shown in Supplementary Fig. 9. We also added a section, “Surface expression”, to the Methods as follows.

a

b

Supplementary Fig. 9 | Measurement of the cell surface expression level of wild-type and mutant HCAs
HEK-293 cells were used to express each construct and the expression levels were measured by ELISA. Data are mean \pm s.e.m. (n=3-4)

Surface expression

Cell-surface expression of HCA2, HCA3, and its mutants were measured by ELISA chemiluminescence. In brief, 48-h post-transfected cells plated in 96 white-well plates were fixed with 50 µl per well 10% (v/v) formaldehyde for 10 min at room temperature. The cells were then washed twice with 80 µl per well of phosphate-buffered saline (PBS) and incubated with 50 µl per well 5% (v/v) BSA in PBS for 1 h. Cells were incubated with an anti-Flag-horseradish peroxidase-conjugated antibody (Sigma-Aldrich, A8592) diluted 1:10,000 in 5% (v/v) BSA in PBS for 1 h at room temperature. After washing 3 times with 80 µl per well PBS, 50 µl per well Super Signal Enzyme-Linked Immunosorbent Assay Pico Substrate (Thermo Fisher, 37070) was added to each well for development of the signal and the luminescence was counted using a SpectraMax i3x multi-plate reader. The luminescence signal was analyzed in GraphPad Prism 9.0 and data were normalized to the signal of wild-type HCA2 (or wild-type HCA3).

3. In line 106-107, “Interestingly, R3.36 is conserved only in the HCA family, including OXER and GPR313 (Supplementary Fig. 9).” This sentence is quite confusing, does it mean OXER and GPR31 belong to the HCA family? Also, I didn’t see GPR31 in Supplementary Fig.9. This should be added.

Response:

We apologize for the unclear sentences. We corrected the sentences and revised the figure showing the multiple sequence alignment in the new Supplementary Fig. 10, as follows.

Page 5, line 121

R^{3.36} is conserved only in the HCA family, OXER, GPR31³, and GPR35 (Supplementary Fig. 10).

Supplementary Fig. 10 | Multiple sequence alignment of class A GPCRs

Sequence alignment of amino acid residues in TM3 of HCA receptors and other class A GPCRs. Residues at position 3.36 are indicated by a red box. Alignment data were obtained from the GPCRdb (gpcrdb.org).

4. In Supplementary Fig.8e, the S179A appeared to improve the EC50 and Emax. While S179 was essential for the ligand binding, what's the possible explanation for this result?

Response:

Thank you for your suggestion. In the MK6892 bound structure, the position of H189 showed a different orientation from the other ligand-bound structures. In these structures except for the MK6892-bound structure, S179 and H189 form hydrogen bonds, but no hydrogen bond was observed in the MK6892-bound structure. The hydrogen bond appears to inhibit MK6892 binding. Therefore, S179A may have increased activity. We added the following text and Supplementary Fig 11.

page 5, line 152

In these structures, except for the MK6892-bound structure, S179 and H189 form hydrogen bonds, but the hydrogen bond was not observed in the MK6892-bound structure and may be rather unnecessary. Consistent with this observation, S179A slightly increases the activity induced by MK6892 (Supplementary Fig 8e). These observations suggest the high flexibility of the ligand-binding pocket and explain the ability of the HCA2 receptor to bind various ligands.

Supplementary Fig. 11 | Indirect interactions essential for ligand pocket formation
a R251^{6.55} and a hydrophobic cluster composed of F180^{ECL2}, F193^{5.43}, and F276^{7.35} in HCA2 are important for stabilizing the ligand binding pocket. The hydrogen bond is indicated by a blue dashed line and the π -cation interaction is indicated by a black dashed line. **b-e** Hydrogen bond between S179 and H189. All ligand and focused residues are shown by the stick model.

5. Since there are no inactive structures for HCAs, SUCR and predicted AF2 model were used for discussing activation mechanisms of HCAs. The conclusions from comparing with AF2 model may not be convincing. Also I was not convinced by the “high sequence similarity” with SUCR. Sequence comparison on the region critical for activation between HCAR and SUCR could be included in the figure to make the conclusion more reliable.

Response:

We thank the reviewer for these comments. We completely agree that the AF2 model may not be convincing. We removed the section on comparison with the AF2 model and replaced it with a comparison with the previously reported δ -branch GPCRs. In addition, we performed a comparison with our active HCA2 structure and the recently reported inactive structure. We added some sentences regarding the activation mechanisms to the Discussion.

page 6, line 179

HCA2 and SUCR are classified in the δ branch of class A GPCRs, including P2Y1³⁹, P2Y12⁴⁰,

CysLT1/2^{41,42}, PAR1/2^{43,44}, PAFR⁴⁵, GPR35³¹, and LPA6⁴⁶. We investigated whether the conformation changes observed in comparison with SUCR are conserved features among class A GPCR family members by comparing the structures of the reported δ -branch GPCRs. Most class A GPCRs have Trp residues at position 6.48 in TM6, which recognize their ligands and initiate the conformational changes required for receptor activation, while the δ -branch receptors have Phe or Tyr residues instead of Trp at this position (Supplementary Fig. 12a). Focused on the conformational change near the microswitch upon agonist binding, the upper half of TM5 of HCA2 bends inward starting at conserved Pro^{5.50}, and upward shifts of TM3 are commonly observed in all the known structures of the δ -branch GPCRs (Supplementary Fig. 12b-i). The rotamer of F^{6.44} and F^{6.48} adopts a “downward” conformation in all the structures, and TM6, including these residues, in the active HCA2 is pushed outward. The conformation observed in HCA2 is also consistent with that of G13-bound GPR35 (Supplementary Fig. 12j), suggesting that other δ -branch receptors have similar activation mechanisms.

Supplementary Fig. 12 | Structural comparison of δ -branch GPCRs focused on the micro-switch

a Sequence alignment of amino acid residues focused on the conserved microswitch of δ -branch class A GPCRs. **B-j** Structural comparison of HCA2 (orange) with other δ -branch class A GPCRs (gray), including PAR1 (PDB: 3VW7), PAR2 (PDB: 5NDD), CysLT1 (PDB: 6RX5), CysLT2 (PDB: 6RZ6), P2Y1 (PDB: 4XNW), P2Y12 (PDB: 4PXZ), PAFR (PDB: 4PXZ), LPA6 (PDB: 5XSZ), and GPR35 (PDB: 8H8J)

page 14, line 301

To gain further insight into the mechanism underlying the receptor activation, we compared our active-state structures with the recently published inactive HCA2 structure. First, a significant difference is that the TM5 helix is positioned more outward in the

inactive structure relative to the cryo-EM structure (Supplementary Fig. 15a). Second, R111^{3.36} and F180^{ECL2}, which are important for ligand binding, undergo major conformational changes. Furthermore, the microswitches (CW(F)xP and PIF motif) show similar changes as in the comparison of δ -branch class A GPCRs. (Supplementary Fig. 15c). Finally, a comparison of the binding interfaces to the C-terminal helix of the Gi subunit revealed a rearrangement of R125^{3.50} (the DRY motif) and Y294^{7.53} (the NPxxY motif) (Supplementary Fig. 15d). We found that F232^{6.36} of TM6 is rotated about 80° relative to the inactive structure. Substitution of the residue remarkably reduced the efficacy, suggesting that F232^{6.36} on the cytoplasmic side of HCA2 is partially responsible for the outward movement of TM6 (Supplementary Fig. 15e). Comparison with the inactive form of HCA2 is comprehensively consistent with our results on the activation mechanism.

Supplementary Fig. 15 | Structural comparison between active and inactive structures of HCA2

a Comparison of the active HCA2 structure with the inactive HCA2 structure (PDB: 7ZLY). Blue arrows indicate differences in the positions of TM5. **b** Close-up view of the extracellular side. Side chains revealing conformational differences between the 2 structures are shown in stick representation. **c** Close-up view of the PIF motif. **d** Close-up view of the cytoplasmic side. Conformational changes of R^{3.50} (DRY motif) and Y^{7.53} (NPxxY motif) show a transition to the activated state. **e** cAMP inhibition assay shows that F232A decreases the efficacy, but not the potency.

6. In the Gi coupling part, it is unclear which structure was used to represent active HCA. Was the Gi coupling interface of four structures exactly the same?

Response:

We apologize for our unclear description. Because the Gi-binding interfaces of the 4 HCA2-Gi structures were almost identical, we used the GSK256073-bound structure with the highest resolution in the Gi coupling part. We added the following sentence:

page 12, line 266

Because the Gi-binding interfaces of the 4 HCA2-Gi structures were almost identical, the GSK256073-bound structure with the highest resolution was used to examine the selectivity in the following section.

7. The discussion on the ligand entrance section was interesting and may provide one important highlight of this paper, however the experimental support to this conclusion is not solid. Since the W188A and H189A mutation experiment did not provide convincing evidence for the ligand entry from the membrane. Was there any other mutations or experiments which can fully block the space between TM4 and TM5?

Response:

We agree that additional evidence for the potential lateral ligand entry would be attractive. Designing mutants that completely block the ligand entrance is difficult, however, because the area around the ligand entrance is already composed of bulky amino acids, and thus we were unable to perform mutational experiments in which the space between TM4 and TM5 is fully blocked. We speculate that the leading factor in closing the ligand entrance is inward bending of the upper half of TM5 upon agonist binding. Importantly, although GSK256073, LUF6283, and acifran occupy the central deep part of the orthosteric pocket, their potency is affected by mutations around the proposed lateral entry site, far away from its final binding site (Supplementary Fig. 15d, now Fig. 4e). Such experiments in which amino acid mutations in the ligand entry reduce ligand affinity have been performed on monoamine receptors as well (Xu et al., 2020). In a recently reported paper, the inactive state structure of HCA2 by constitutive inactive mutation was analyzed (Yang et al, 2023). Compared with our activated HCA2 structure, the space between TM4 and TM5 appeared to be open, potentially allowing ligands to enter through the entrance and reach the orthosteric pocket. We added these details to the Discussion as follows.

page 14, line 301

To gain further insight into the mechanisms underlying the receptor activation, we compared our active-state structures with the recently published inactive HCA2 structure. First, a significant difference is that the TM5 helix is positioned more outward in the inactive structure relative to the cryo-EM structure (Supplementary Fig. 15a). Second, R111^{3.36} and F180^{ECL2}, which are important for ligand binding, undergo major conformational changes. Furthermore, the microswitches (CW(F)xP and PIF motif) show similar changes as in the comparison of δ -branch class A GPCRs. (Supplementary Fig. 15c). Finally, a comparison of the binding interfaces to the C-terminal helix of the Gi subunit revealed a rearrangement of R125^{3.50} (the DRY motif) and Y294^{7.53} (the NPxxY motif) (Supplementary Fig. 15d). We found that F232^{6.36} of TM6 is rotated about 80° relative to the inactive structure. Substitution of the residue remarkably reduced the efficacy, suggesting that F232^{6.36} on the cytoplasmic side of HCA2 is partially responsible for the outward movement of TM6 (Supplementary Fig. 15e). Comparison with the

inactive form of HCA2 is comprehensively consistent with our results on the activation mechanism.

Supplementary Fig. 15 | Structural comparison between active and inactive structures of HCA2

a Comparison of the active HCA2 structure with the inactive HCA2 structure (PDB: 7ZLY). Blue arrows indicate differences in the positions of TM5. **b** Close-up view of the extracellular side. Side chains revealing conformational differences between the 2 structures are shown in stick representation. **c** Close-up view of the PIF motif. **d** Close-up view of the cytoplasmic side. Conformational changes of R^{3.50} (DRY motif) and Y^{7.53} (NPxxY motif) show a transition to the activated state. **e** cAMP inhibition assay shows that F232A decreases the efficacy, but not the potency.

8. The scale bars in Supplementary Fig. 2b&e, 3b&e, 4b should be indicated for their length.

Response:

Thank you for this suggestion. We added the scale bar lengths to the Supplementary Figure legends as follows.

Representative micrograph and 2D class averages (scale bars indicate 20 nm and 5 nm, respectively)

9. Line 129 in Supplementary Fig.15 legend, I think the “antagonist” should be “agonist”.

Response:

Thank you for pointing this out. Yes, we meant “agonist” and this has been corrected.

10. The whole manuscript needs more work to polish the text. Due to grammatical errors and unclear logical structure, some sentences are difficult for readers to understand. Just to give one example in line 45-46: HCA2 was first identified as a receptor for niacin and is the most well-studied pharmacologically. Also, from line 251-256, the N-terminus beta-hairpin and C18-C19 disulfide bond repeated twice.

Response:

The whole manuscript was polished, and grammatical errors and unclear logical structures were reviewed and corrected. Please see the revised manuscript, as we have corrected a great many parts of the text.

Reviewer #4 (Remarks to the Author):

This manuscript describes structural analyses of agonist occupied hydroxycarboxylic acid receptor subtype 2 and 3 together with mutagenesis, binding/activation data and molecular dynamics simulation. The work provide molecular insight into ligand binding and activation of HCA2 and HCA3 receptors and close family member. The authors selected to solve the cryoEM structures of human HCA2 heterotrimeric Gi complexes bound to the selective full agonist GSK256073, MK6892, partial agonist LUF6283 and non-selective full agonist acifran. To understand further agonist binding/activation mechanism, the acifran was also solved with complexes of human HCA3 heterotrimeric Gi. The structural analysis was complemented extensive mutation analysis of agonists binding and probe Gi protein activation using functional assays and molecular dynamics.

The work is carefully performed, well presented, and provides useful and noteworthy information for the field of GPCRs and short-chain fatty acid signalling receptors, and their ligand binding interactions, and more generally their potency as therapeutic targets towards dyslipidaemia, niacin flush and other autoimmune diseases. The work is very well suited for wide audience of Nature communications.

I have no major comments flaws in the data analysis, interpretation and conclusions regarding this manuscript. The methodology meets the standard usage in structural biology, the produced structures meet the validation criteria, data binding/functional data related mutagenesis and interpretation is well preformed, and sufficient detail are provided in the methods for the work to be reproduced.

Response:

We thank the reviewer for these positive comments on our work.

Minor points

Active conformation of HCAs, lines 157-178

It is not clear to me, how to results suggest that HCAs have unique regulation mechanism among Gi/o coupled reported so far. What is described is very similar binding mode for Gi/o. Maybe rewrite and clarify the paragraph. The lower part of paragraph could be joined the next paragraph "Gi coupling interface"

Response:

We thank the reviewer for these comments. We rewrote and reorganized this paragraph as suggested. We now describe it as unique regulation based on the following 2 observations. First, the movement of TM6 in HCA2 was significantly less than that in other Gi-binding GPCRs. Second, ICL2 in HCA2 is hardly involved in the Gi binding. This, however, was an overstatement. First, we avoided the word "unique" with regard to the Gi coupling. Second, in response to Reviewer #1 (comment 1), the Gi coupling section was shortened to clarify the main topic of the manuscript. We revised the manuscript as follows.

page 12, line 264

Gi coupling interface

While HCA2 is reported to couple with only Gi, no clear mechanism for the selectivity has been reported. Because the Gi-binding interfaces of four HCA2-Gi structures were almost identical, the GSK256073-bound structure with the highest resolution was used to examine

the selectivity in the following section. Our structures indicate that the binding of HCA2 to Gi uses 4 interfaces (Fig. 6a). The first is between the cytoplasmic ends of TM3, TM5, and TM6 in HCA2, and the C-terminal $\alpha 5$ helix of Gi. V129^{3,54} of TM3, I211^{5,61}, and I215^{5,65} of TM5, and I233^{6,37} and I226^{6,30} of TM6 in HCA2 form hydrophobic interactions with L344, L348, and L353 of the Gi subunit (Fig. 6b). The hydrophobic interaction is a common feature of other Gi-bound GPCRs, including dopamine receptor 3⁸ and cannabinoid receptor type 1⁴. The second is between intracellular loop 2 (ICL2) and the hydrophobic cleft composed of Gi (Fig.6c), which is also observed in Gs- and Gi-coupled GPCRs. In general, hydrophobic residues are conserved at position 34.51. In HCA1-3, a histidine residue is located at this position and interacts with αN , the $\beta 2$ - $\beta 3$ loop, and $\alpha 5$ in Gi to fit into the hydrophobic groove formed by L194, F336, and I343. The third is the interaction between ICL3 of HCA2 and the G protein. ICL3 of HCA2 is very short, but residues R218^{ICL2} and R222^{ICL2} in the receptor form a polar interaction with D337/D341 and E318 in Gi, respectively (Fig. 6e). Fourth, K57 of ICL1 interacts with D350 (Fig. 6d). We introduced mutations into these amino acids and measured the Gi activity (Fig. 6f, g). Only R218A exhibited remarkably decreased Gi activity. The corresponding residues in other Gi-coupled GPCRs form polar interactions with D341 in Gi, suggesting a contribution to G protein selectivity^{7,8,21}. R128^{3,47}, R222^{ICL3}, and H223^{ICL3} are not critical for specific interactions with Gi. These residues may be required, however, for the positively charged property of these parts of the cytoplasmic side, which is one of the key driving forces for coupling GPCRs and Gi proteins^{8,22}.

Fig. 6 | The G protein interfaces of HCA2

a Interface between HCA2 (cartoon representation) and Gi-protein (surface representation). **b** Hydrophobic interactions between HCA2 and the C terminus of the Gi subunit. **c** Interactions between ICL2 and a hydrophobic cleft formed by Gi. **d** Interactions of ICL3 with Gi. **e** Interactions of ICL1 with the C-terminus of Gi. Hydrophilic interactions are indicated by blue dashed lines. **f** Close-up view of the interface between HCA2 and Gi. Side chains of positively and negatively charged amino acids are shown in stick representation. The electrostatic potential surface of Gi ranges from -10 kT/e (red) to $+10\text{ kT/e}$ (blue). **g** cAMP inhibition assay of Gi protein-binding interface mutants.

The authors constructed inactive model of HCA2 using alphafold and compared it to active-state to further understand the helical changes during activation. Maybe using experimentally

determined inactive-state structures of succinate or other close by family members could be used instead of the alphafold model?

Response:

We thank the reviewer for these comments. According to the reviewer's suggestions, we compared several δ -branch GPCRs and added them as a new figure (Supplementary Fig. 12). The discussion of AF2 was removed from the manuscript, and we added a comparison with the recently reported crystal structures of HCA2 in an inactive state to the Discussion as follows, together with the new Supplementary Figure 15 as mentioned above in the response to Reviewer #3 (comment 7).

page 9, line 178

HCA2 and SUCR are classified in the δ branch of class A GPCRs, including P2Y1³⁹, P2Y12⁴⁰,

CysLT1/2^{41,42}, PAR1/2^{43,44}, PAFR⁴⁵, GPR35³¹, and LPA6⁴⁶. We investigated whether the conformation changes observed in comparison with SUCR are conserved features among class A GPCR family members by comparing the structures of the reported δ -branch GPCRs. Most class A GPCRs have Trp residues at position 6.48 in TM6, which recognize their ligands and initiate the conformational changes required for receptor activation, while the δ -branch receptors have Phe or Tyr residues instead of Trp at this position (Supplementary Fig. 12a). Focused on the conformational change near the microswitch upon agonist binding, the upper half of TM5 of HCA2 bends inward starting at conserved Pro^{5.50}, and upward shifts of TM3 are commonly observed in all the known structures of the δ -branch GPCRs (Supplementary Fig. 12b-i). The rotamer of F^{6.44} and F^{6.48} adopts a "downward" conformation in all the structures, and TM6, including these residues, in the active HCA2 is pushed outward. The conformation observed in HCA2 is also consistent with that of G13-bound GPR35 (Supplementary Fig. 12j), suggesting that other δ -branch receptors have similar activation mechanisms.

Supplementary Fig. 12 | Structural comparison of δ -branch GPCRs focused on the micro-switch

a Sequence alignment of amino acid residues focused on the conserved microswitch of δ -branch class A GPCRs. **B-j** Structural comparison of HCA2 (orange) with other δ -branch class A GPCRs (gray), including PAR1 (PDB: 3VW7), PAR2 (PDB: 5NDD), CysLT1 (PDB: 6RX5), CysLT2 (PDB: 6RZ6), P2Y1 (PDB: 4XNW), P2Y12 (PDB: 4PXZ), PAFR (PDB: 4PXZ), LPA6 (PDB: 5XSZ), and GPR35 (PDB:8H8J)

page 14, line 302

To gain further insight into the mechanism underlying the receptor activation, we compared our active-state structures with the recently published inactive HCA2 structure. First, a significant difference is that the TM5 helix is positioned more outward in the

inactive structure relative to the cryo-EM structure (Supplementary Fig. 15a). Second, R111^{3.36} and F180^{ECL2}, which are important for ligand binding, undergo major conformational changes. Furthermore, the microswitches (CW(F)xP and PIF motif) show similar changes as in the comparison of δ -branch class A GPCRs. (Supplementary Fig. 15c). Finally, a comparison of the binding interfaces to the C-terminal helix of the Gi subunit revealed a rearrangement of R125^{3.50} (the DRY motif) and Y294^{7.53} (the NPxxY motif) (Supplementary Fig. 15d). We found that F232^{6.36} of TM6 is rotated about 80° relative to the inactive structure. Substitution of the residue remarkably reduced the efficacy, suggesting that F232^{6.36} on the cytoplasmic side of HCA2 is partially responsible for the outward movement of TM6 (Supplementary Fig. 15e). Comparison with the inactive form of HCA2 is comprehensively consistent with our results on the activation mechanism.

Supplementary Fig. 15 | Structural comparison between active and inactive structures of HCA2

a Comparison of the active HCA2 structure with the inactive HCA2 structure (PDB: 7ZLY). Blue arrows indicate differences in the positions of TM5. **b** Close-up view of the extracellular side. Side chains revealing conformational differences between the 2 structures are shown in stick representation. **c** Close-up view of the PIF motif. **d** Close-up view of the cytoplasmic side. Conformational changes of R^{3.50} (DRY motif) and Y^{7.53} (NPxxY motif) show a transition to the activated state. **e** cAMP inhibition assay shows that F232A decreases the efficacy, but not the potency.

REVIEWER COMMENTS

Reviewer #1 (Remarks to the Author):

Most of my comments have been addressed.

However there still some remaining outstanding issues with the models:

The GSK256073 still doesn't have planarity around the N atom and it is quite noticeable even in the provided "figure 1 for response". I am taking about the "C N C1 C4" plane.

The numbers of favoured rotamers continues to hover around 70-80% (checked for 8IHK and 8IHJ only). While this doesn't affect the conclusions it is quite important to deposit the correct coordinates.

Reviewer #3 (Remarks to the Author):

I think the revisions addressed my major concerns. However I noticed that in the rebuttal letter, response to my question #7 is partially overlapped with that to question #5. Given my concern in #7 is related to the ligand entry path, a major highlight of this manuscript in my view, I expect to see a more structured revision to this specific question. I think this is the only piece of remaining concerns before I can agree with the publication.

Reviewer #4 (Remarks to the Author):

The authors have addressed all my and other reviewer's comments for this manuscript. The manuscript has been significantly improved after revising. Therefore, I have no further comments and I recommend accepting the manuscript for publication.

Point-to-Point Response to the Reviewers

Manuscript ID; NCOMMS-23-08315

We sincerely appreciate the valuable and insightful comments of the editor and reviewers on our manuscript. We have revised and improved our manuscript as suggested, and provide a point-by-point response to address each comment and question. Parts of the manuscript that have been updated in response to the reviewers' comments are highlighted in the text.

REVIEWER COMMENTS

Reviewer #1 (Remarks to the Author):

Response:

We thank the reviewer for these positive and very clear specific comments and for the valuable advice on PDB models.

Most of my comments have been addressed.

However there still some remaining outstanding issues with the models:

The GSK256073 still doesn't have planarity around the N atom and it is quite noticeable even in the provided "figure 1 for response". I am taking about the "C N C1 C4" plane.

Response:

Thank you for your comment. We apologize for the insufficient refinement of the compound model. We have corrected the planarity of C N C1 C4 according to the reviewer's suggestion, as shown in the figure below.

Figure 1 for revision; Ligand and density map of GSK256073

Superimposed ligand and density map. GSK256073 is shown in stick representation. cryo-EM density is represented as a transparent surface.

The numbers of favoured rotamers continues to hover around 70-80% (checked for 8IHK and 8IHJ only). While this doesn't affect the conclusions it is quite important to deposit the correct coordinates.

Response:

We appreciate you checking the model in very detail. We have revisited and confirmed all of the models show sufficiently high numbers of favoured rotamers (99.58-99.89%), as shown in the statistical tables below. We believe our models have been adequately built according to the reviewer's suggestions.

	HCA2- GSK256 073 (EMDB- 3542) (PDB 8IHB)	HCA2- MK6892 (EMDB- 35443) (PDB 8IHF)	HCA2- LUF6283 (EMDB- 35444) (PDB 8IHH)	HCA2- acifran (EMDB- 35445) (PDB 8IHI)	HCA3- acifran (EMDB- 35446) (PDB 8IHJ)	HCA3- acifran (local) (EMDB -35447) (PDB 8IHK)
Data collection and processing						
Magnification	105K	105K	50K	50K	60K	
Voltage (kV)	300	300	300	300	300	
Electron exposure (e ⁻ / Å ²)	49	49	69.6	69.6	71.2	
Defocus range (µm)	-0.7 to - 1.5	-0.7 to - 1.5	-1.0 to -2.0	-1.0 to - 2.0	-1.0 to - 2.0	
Pixel size (Å)	0.83	0.83	0.99	0.99	0.78	
Final particle images (no.)	245,517	181,958	181,273	146,577	95,567	
Map resolution (Å)	2.85	2.97	3.13	3.17	3.18	3.33
FSC threshold	0.143	0.143	0.143	0.143	0.143	0.143
Map resolution range (Å)	2.8-4.8	2.8-4.8	2.8-4.8	2.8-4.8	2.8-4.8	2.8-4.8
Refinement						
Model resolution (Å)	3.0	3.1	3.2	3.2	3.3	3.6
FSC threshold	0.5	0.5	0.5	0.5	0.5	0.5
Model resolution range (Å)	n/a	n/a	n/a	n/a	n/a	n/a
Map sharpening B factor (Å ²)	-96.8	-97.4	-87.8	-87.3	-94.5	-100.4
Model composition						
Non-hydrogen atoms	8709	8702	8649	8672	8510	2253
Protein residues	1131	1131	1131	1131	1122	283
Ligands	1	1	1	1	1	1
R.m.s. deviations						
Bond lengths (Å)	0.004	0.003	0.002	0.003	0.002	0.002
Bond angles (°)	0.6151	0.495	0.519	0.522	0.463	0.509
Validation						
MolProbity score	1.59	1.59	1.60	1.44	1.50	1.37
Clashscore	6.50	5.98	6.35	5.72	5.55	3.80
Poor rotamers (%)	0.11	0.11	0.11	0.22	0.11	0.00
Ramachandran plot						
Favored (%)	96.50	96.14	96.32	97.31	96.75	96.80
Allowed (%)	3.50	3.86	3.68	2.69	3.25	3.20
Disallowed (%)	0	0	0	0	0	0

Supplementary Table 1

Cryo-EM data collection, refinement, and validation statistics

Reviewer #3 (Remarks to the Author):

Response:

We thank the reviewer for his or her constructive comments and valuable feedback. We are pleased to have addressed many of the reviewer's concerns. I will be happy to address the remaining concerns.

I think the revisions addressed my major concerns. However I noticed that in the rebuttal letter, response to my question #7 is partially overlapped with that to question #5. Given my concern in #7 is related to the ligand entry path, a major highlight of this manuscript in my view, I expect to see a more structured revision to this specific question. I think this is the only piece of remaining concerns before I can agree with the publication.

Response:

I apologize for the inaccurate response. Perhaps the most important question raised by the reviewer concerns the lateral entry pathway through lipid membranes. Although our mutant analysis and MD simulation clearly suggest a ligand entry site, we cannot provide definitive evidence for the lateral entry pathway through lipid membranes. Therefore, we have decided to avoid making claims such as lateral entry from within the membrane in this paper.

We changed the manuscript as follows.

page,10 line,209

The following sentences have been removed.

” This means that in contrast to receptors that bind ligands from the extracellular side, such as amine-coupled receptors, the ligand entry to the binding pockets of HCA2 would be from the lipid membrane through the lateral gate.”

page,10 line 213

We changed the sentence ”Considering the extended ligand-binding pocket in the MK6892-bound structure, we hypothesize that the gap between TM4 and TM5 is the entrance to the ligand pocket” to” Considering the extended ligand-binding pocket in the MK6892-bound structure, we hypothesize that the gap between TM4 and TM5 and ECL2 is the entrance to the ligand pocket”.

page,11 line,225

We changed the sentence “These results suggest that the space between TM4 and TM5 is the ligand entrance in HCA2, and endogenous agonists and synthetic agonists insert into the ligand pocket from the lateral gate within the lipid bilayer membrane.” to” These results suggest that the space between TM4 and TM5 and ECL2 is the ligand entrance in HCA2, and endogenous agonists and synthetic agonists insert through the flexible open/close entrance and are occluded in the orthosteric binding pocket.”

page, 14 line,298

We changed the sentence “lateral gate between TM4 and TM5” to “the gap between TM4 and TM5 and ECL2”.

Reviewer #4 (Remarks to the Author):

The authors have addressed all my and other reviewer's comments for this manuscript. The manuscript has been significantly improved after revising. Therefore, I have no further comments and I recommend accepting the manuscript for publication.

Response:

We are pleased to hear that all comments have been addressed. We also greatly appreciate the reviewer for all the constructive comments and suggestions.

REVIEWERS' COMMENTS

Reviewer #1 (Remarks to the Author):

All of my comments have been addressed. I very much recommend this manuscript for publication in Nature Communications.

Point-to-Point Response to the Reviewers

Manuscript ID; NCOMMS-23-08315

We sincerely appreciate the editor's and reviewers' valuable and insightful comments on our manuscript.

REVIEWER COMMENTS

Reviewer #1 (Remarks to the Author):

All of my comments have been addressed. I very much recommend this manuscript for publication in Nature Communications.

Response:

We are pleased to hear that all comments have been addressed. We also greatly appreciate the reviewer for all the constructive comments and suggestions on PDB model validation.